# Large Language Models Are Not Robust Multiple Choice Selectors

**Chujie Zheng**[†]    **Hao Zhou**[‡]    **Fandong Meng**[‡]    **Jie Zhou**[‡]    **Minlie Huang**[†*]

[†]The CoAI Group, DCST, BNRist, Tsinghua University, Beijing 100084, China
[‡]Pattern Recognition Center, WeChat AI, Tencent Inc., China
chujiezhengchn@gmail.com    aihuang@tsinghua.edu.cn

## Abstract

Multiple choice questions (MCQs) serve as a common yet important task format in the evaluation of large language models (LLMs). This work shows that modern LLMs are vulnerable to option position changes in MCQs due to their inherent "selection bias", namely, they prefer to select specific option IDs as answers (like "Option A"). Through extensive empirical analyses with 20 LLMs on three benchmarks, we pinpoint that this behavioral bias primarily stems from LLMs' token bias, where the model a priori assigns more probabilistic mass to specific option ID tokens (e.g., A/B/C/D) when predicting answers from the option IDs. To mitigate selection bias, we propose a label-free, inference-time debiasing method, called PriDe, which separates the model's prior bias for option IDs from the overall prediction distribution. PriDe first estimates the prior by permuting option contents on a small number of test samples, and then applies the estimated prior to debias the remaining samples. We demonstrate that it achieves interpretable and transferable debiasing with high computational efficiency. We hope this work can draw broader research attention to the bias and robustness of modern LLMs.[1]

## 1 Introduction

Multiple choice question (MCQ) is a prevalent input format of large language models (LLMs). An MCQ typically encompasses a question accompanied by multiple candidate options, from which the model is tasked to select the most suitable answer, as exemplified in Figure 1. Current LLM-centric scenarios have widely utilized the task format of MCQ, for instance, within benchmarks targeted at assessing LLMs (Hendrycks et al., 2020; Zhong et al., 2023; Huang et al., 2023) and in LLM-based automatic evaluation frameworks (Chiang et al., 2023; Zheng et al., 2023b). In any scenario, we always expect LLMs to robustly select reliable answers in MCQs.

```
Question:  In an SR latch
built from NOR gates, which
condition is not allowed
Options:
A. S=0, R=0   B. S=0, R=1
C. S=1, R=0   D. S=1, R=1
Answer:  D
```

Figure 1: A multiple choice question (MCQ) example from MMLU.

Unfortunately, we observe that *modern LLMs are vulnerable to option position changes in MCQs* (Robinson & Wingate, 2022). We show in Table 1 that, in the 0-shot MMLU evaluation (Hendrycks et al., 2020), a simple "answer-moving attack" by always moving the golden answers to a specific position causes LLMs' dramatic performance fluctuations. For instance, moving the golden answers to position D degrades the accuracy of gpt-3.5-turbo by 6.3 (from 67.2 to 60.9). When moving to A, llama-30b is boosted by 15.2 and surpasses gpt-3.5-turbo (68.2 vs. 65.3), which starkly contrasts with their original performance (53.1 vs. 67.2).

LLMs' poor robustness to option position changes results from their biased behavior: *they prefer to select specific option IDs as answers* (like "Option A"), which we term as **selection bias**. As a simple verification, we randomly sampled 1,000 MMLU test samples, where we controlled the number of correct answers being A/B/C/D as 250 each. Among these samples, llama-30B selects A/B/C/D 34.6% / 27.3% / 22.3% / 15.8% of the time, while gpt-3.5-turbo for 22.5% / 25.6% /

---

[*]Corresponding author.
[1]Project repository: https://github.com/chujiezheng/LLM-MCQ-Bias.

Table 1: Simply moving the golden answers of MCQs to a specific option position can cause dramatic performance fluctuations (0-shot MMLU), suggesting LLMs' poor robustness to option position changes in MCQs.

| Move Golden to | Orig | A | B | C | D |
|---|---|---|---|---|---|
| `llama-30B` | 53.1 | 68.2 (+15.2) | 54.1 (+1.1) | 50.1 (-2.9) | 41.2 (-11.9) |
| `vicuna-v1.3-33B` | 57.0 | 59.5 (+2.5) | 58.6 (+1.5) | 65.8 (+8.8) | 44.8 (-12.3) |
| `falcon-40B` | 51.8 | 46.3 (-5.5) | 45.2 (-6.7) | 64.8 (+13.0) | 47.9 (-3.9) |
| `falcon-inst-40B` | 51.5 | 38.3 (-13.3) | 38.9 (-12.7) | 55.7 (+4.1) | 69.1 (+17.6) |
| `llama-2-70B` | 64.0 | 61.5 (-2.6) | 68.6 (+4.5) | 64.1 (+0.1) | 62.0 (-2.1) |
| `gpt-3.5-turbo` | 67.2 | 65.3 (-1.9) | 68.5 (+1.3) | 74.2 (+6.9) | 60.9 (-6.3) |

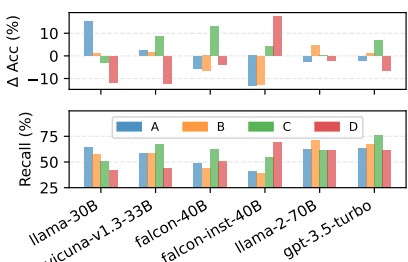

Figure 2: The performance fluctuation (left) correlates with the balance of different option IDs' recalls in the original inputs, as *the original recall of option ID X is actually calculated on a subset of samples where all the golden answers are moved to X.*

32.3% / 19.6%, respectively (averaged over 10 runs). These proportions are statistically nonuniform ($\chi^2$ Test, $p$-value $\ll 10^{-4}$) and align well with the performance fluctuations in Table 1.

Through extensive empirical evaluation (§2.3), with 20 LLMs, on three benchmarks, and with varying option numbers (from two to five), we show that selection bias is prevalent across various LLMs and cannot be well mitigated by simple prompting strategies (§2.6), like Chain-of-Thought prompting (Wei et al., 2022; Kojima et al., 2022). It varies with models but manifests a cross-domain similarity within the same model (§2.3). With careful ablation analyses (§2.4), we find that, contrary to the common view in previous work (Wang et al., 2023a; Pezeshkpour & Hruschka, 2023), selection bias arises less from LLMs' position bias, where they are deemed to favor options presented at specific ordering positions (like first or last). In contrast, we pinpoint one more salient intrinsic cause of selection bias as the model's **token bias** when predicting answers from the option IDs given the standard MCQ prompt, where *the model a priori assigns more probabilistic mass to specific ID tokens* (e.g., `A`/`B`/`C`/`D`).

To efficiently mitigate selection bias, we propose a method called **PriDe** (§3), referring to **De**biasing with **Pri**or estimation. In PriDe, we assume the model's prior bias for option IDs can be separated from the overall prediction distribution. PriDe first estimates the prior by permuting option contents on a small number of test samples (e.g., 5%), and then applies it to debias the remaining samples. The whole debiasing procedure needs no sample labels, takes place during the inference time, and requires only negligible extra computational costs, which is especially suitable for modern LLMs. We demonstrate that PriDe achieves superior debiasing effectiveness to strong baselines, especially in the low-cost scenario (§4.1). Furthermore, the prior estimated by PriDe provides a good interpretation for selection bias (§4.1) and can transfer well across different domains (§4.2), which highlights its practical potential in broader scenarios.

**Summary of Contributions** (1) We identify the ubiquitous selection bias in LLMs and provide extensive empirical analyses (with 20 LLMs, on three MCQ benchmarks) and valuable insights on this problem. (2) We pinpoint LLMs' token bias as one primary intrinsic cause of selection bias. (3) We propose a label-free, inference-time debiasing method PriDe, which demonstrates notable effectiveness and efficiency, interpretability, and cross-domain transferability. We hope this work can inspire future research on the bias and robustness of LLMs.

## 2 INVESTIGATION ON SELECTION BIAS

### 2.1 EXPERIMENTAL SETUP

**Models** Our study focuses on the causal, decoder-only LLMs since this architecture has become the dominant choice for modern LLMs. We experiment with *20 LLMs from popular LLM families across various sizes*: `llama-7/13/30/65B` (Touvron et al., 2023a), `llama-2(-chat)-7/13/70B` (Touvron et al., 2023b), `vicuna-v1.3-7/13/33B`, `vicuna-v1.5-7/13B` (Chiang et al., 2023), `falcon(-inst)-7/40B` (Almazrouei et al., 2023), and `gpt-3.5-turbo-0613` (OpenAI, 2022). The models except `gpt-3.5-turbo` are all open source on the HuggingFace website, and we can

access their output probabilities. `gpt-3.5-turbo` is the commercial API of ChatGPT. It accepts textual prompts and returns generated texts without providing access to output probabilities.

**Benchmarks**  We conduct experiments on MMLU (Hendrycks et al., 2020), ARC-Challenge (Clark et al., 2018), and CommonsenseQA (CSQA) (Talmor et al., 2019), which are all MCQ benchmarks widely used for LLM evaluation. Our selection of benchmarks takes into account the diversity of tasks and domains. Specifically, MMLU and ARC consist of 4-option MCQs, while CSQA consists of 5-option ones, and MMLU covers tests from 4 domain categories spanning 57 subjects. The diverse domains facilitate us to derive general observations and enable cross-domain transfer exploration. See Appendix E for detailed data statistics.

**Evaluation**  Our evaluation protocol *follows the mainstream LLM evaluation frameworks*, such as HuggingFace LLM Leaderboard, EleutherAI lm-harness, the original MMLU implementation, and OpenAI Evals (see Appendix F). Specifically, for open-source models, we access the output probabilities of option ID tokens `A/B/C/D/E` and use the maximal one as the model prediction. For `gpt-3.5-turbo`, we compare the golden answer with the first generated token, with the decoding temperature set to 0. See Figure 7 and 6 in Appendix A for the input formats.

Our evaluation mainly considers the 0-shot setting, which excludes biases introduced by in-context examples, but we also conduct 5-shot experiments. The in-context examples come from the development sets and are shared across all the test samples within the same task.

## 2.2 Measurement of Selection Bias

In our study, selection bias is defined as *the model's behavioral bias to select specific option IDs as answers*. To measure selection bias, one naive way is based on the counting for model predictions, which, however, is susceptible to label imbalance. We instead propose to measure selection bias based on the **balance of recalls** of different option IDs and use **the standard deviation of recalls (RStd)** as a quantitative metric. This measurement is intuitive that greater recall imbalance indicates more pronounced selection bias and is not as susceptible to label imbalance as the counting-based measurement. More importantly, recall balance well reflects the model's robustness to option position changes, as illustrated in Figure 2. Hence, we reasonably expect that *reducing selection bias (measured by recall balance) will improve LLMs' robustness to option position changes in MCQs*.

Note that measuring selection bias with recall balance implies the premise that the golden answers should be randomly placed. We show in Figure 18 in Appendix C that randomly shuffling the options does not obviously change selection bias, validating the above premise.

## 2.3 Key Observations

We first conduct an extensive evaluation of LLMs on various benchmarks to gain a preliminary understanding of selection bias. We show partial results in Figure 3 for a brief presentation and put the full results in Appendix B. We draw the following main observations and insights:

**Selection bias is prevalent across various LLMs and varies with model families and sizes.**  Intuitively, selection bias is likely to originate from LLMs' training data, where some answers (e.g., `C`) may occur more frequently than others. However, we do not observe consistent patterns of selection bias within the same model family where the models are trained with the same training data (e.g., `llama-7/13/30/65B`, `llama-2-7/13/70B`). We speculate that selection bias arises as a product of complex interactions between training data composition and ordering, model capacity (number of parameters), and other factors like hyperparameters.

**Selection bias within the same model displays a moderate similarity across different domains.**
For instance, under the 0-shot setting, `llama-30B` consistently prefers A/B on various benchmarks, while `gpt-3.5-turbo` favors C/B more. While the preference ranking may not strictly persist across tasks or domains, there is an overarching tendency for each model to lean towards certain option IDs (e.g., A and B) and away from others (e.g., C and D). It suggests that *selection bias is an inherent behavioral bias of LLMs that is less impacted by tasks or domains*.

**In-context examples can reduce but may meanwhile alter selection bias.**  As exemplified in Figure 3, `llama-30B` disfavors C under the 0-shot setting but becomes biased towards it under the 5-shot setting. We find that this alteration still does not display noticeable patterns within the same

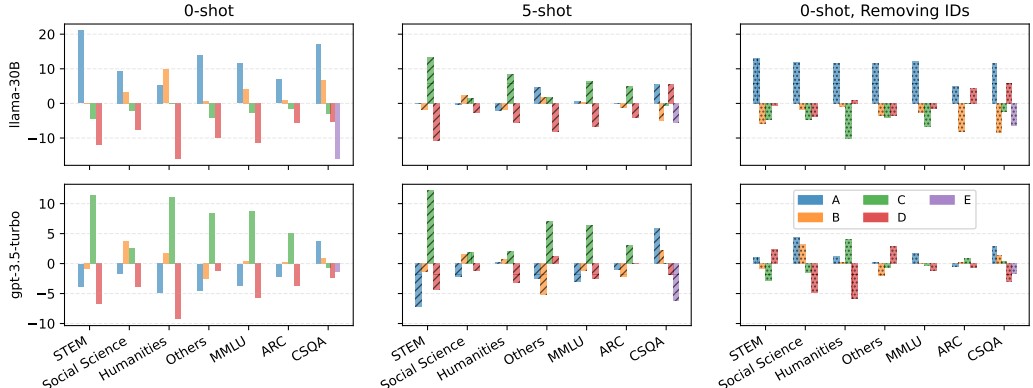

Figure 3: Selection bias of `llama-30B` and `gpt-3.5-turbo` on various benchmarks, see Appendix B for all the 20 LLMs' results. Y-axis: the recall score (%) normalized by *subtracting the overall accuracy*. MMLU is additionally split into four domains (STEM, Social Science, Humanities, Others) based on its subject categories.

model family. It indicates that in-context examples can introduce new biases that will be intertwined with the inherent selection bias, making the latter complex and less regular.

## 2.4 WHAT CAUSES SELECTION BIAS?

Given the ubiquity of selection bias in various LLMs, we now seek to figure out the intrinsic causes resulting in this behavioral bias. We propose two hypotheses: (1) **Token bias**. In the standard MCQ prompt (Figure 1), when selecting answers from the option IDs, the model may *a priori assign more probabilistic mass to specific ID tokens* (such as A or C). (2) **Position bias**. The model may *favor options presented at specific ordering positions* (such as the first or last one). Note that in a recent work, Wang et al. (2023a) similarly found that GPT-4 exhibits the bias towards the responses from "`Assistant 1`". However, it is still unclear whether it is because the preferred responses are "selected via the ID token 1" or because they are "presented first".

One challenge here is that *option IDs are bound with options' ordering positions*, e.g., the ID B is naturally tied with the second-presented option. To distinguish the impacts of the two hypothesized causes, we conduct two ablation experiments. (1) **Shuffling option IDs**. We randomly shuffle the default ID ordering A/B/C/D, for instance, into B/D/A/C or C/A/D/B, etc. In this way, B can denote the option presented at any ordering position, thus eliminating the impact of position bias and leaving only token bias. *Note that this ablation will obviously impair the naturalness and quality of the MCQ prompt, and may consequently degrade model performance (as shown in Table 2).*

Table 2: Preliminary debiasing results (0-shot, `gpt-3.5-turbo`), measured by the standard deviation of recalls (RStd) and accuracy (Acc).

| Methods | MMLU | | ARC | |
|---|---|---|---|---|
| | RStd | Acc | RStd | Acc |
| Default | 5.5 | 67.2 | 3.3 | 84.3 |
| a/b/c/d | 6.8 | 67.0 | 2.1 | 83.1 |
| 1/2/3/4 | 3.8 | 65.8 | 2.1 | 82.3 |
| (A)/(B)/(C)/(D) | 8.1 | 66.5 | 4.0 | 82.4 |
| **Debiasing Instruct** | 6.1 | 66.3 | 3.9 | 84.2 |
| **Chain-of-Thought** | 4.5 | 66.8 | 3.4 | 84.5 |
| **Shuffling IDs** | 5.1 | 63.9 | 3.7 | 80.3 |
| **Removing IDs** | **1.0** | **66.7** | **0.6** | **84.9** |

(2) **Removing option IDs** and asking the model to directly select option contents. In this way, the change of selection bias would indicate the impact of token bias, while the remaining part corresponds to position bias. When evaluating LLMs without option IDs, we require `gpt-3.5-turbo` to generate the whole selected option, which is then compared with the golden answer. For open-source models, we compute the likelihoods of options, normalized by their lengths, and use the maximum one as the model prediction. See Figure 9 and 8 in Appendix A for the input formats.

As shown in Figure 3 and Table 2, the removal of option IDs notably reduces selection bias (RStd decreases), while RStd is little changed by shuffling option IDs. The former observation, in most cases, *holds for various LLMs, on different benchmarks, and with varying option numbers* (from two to five, where 2/3-option settings are constructed from the original data), see Appendix B and Table 3 in Appendix C for detailed results. We also try to replace the default ID symbols A/B/C/D with several reasonable alternatives, including a/b/c/d, 1/2/3/4, and (A)/(B)/(C)/(D), but observe no remarkable reduction in RStd from the default one, as shown in Table 2. These results confirm that *the model's token bias is one primary intrinsic cause of selection bias*.

However, with option IDs removed, the remaining selection bias (corresponding to the impact of position bias) varies with models and tasks, see Appendix B and Table 3 in Appendix C. For instance, the remaining selection bias of `llama-13B`, `vicuna-v1.3-7B`, and `gpt-3.5-turbo` is only marginal, while that of `llama-30B`, `vicuna-v1.3-33B`, and `falcon-40B` is still pronounced (although having been reduced much). The selection bias of `llama-2-13/70B` even slightly increases in MMLU and ARC after option IDs being removed while still decreasing in CSQA, implying the potential counteraction between token bias and position bias. These results suggest that *the model's position bias is somewhat present but quite irregular, largely depending on models and tasks*.

## 2.5 Can We Debias LLMs by Removing Option IDs?

Despite the notably reduced selection bias, we find that removing option IDs usually degrades model performance (except in a few cases under the 5-shot setting), see Table 3 and 4 in Appendix C. This performance degradation results from the way we leverage LLMs to answer MCQs without option IDs, i.e., calculating and comparing the likelihoods of options, which is referred to as the "cloze prompt" format in Robinson & Wingate (2022). Their study demonstrates that asking LLMs to predict option IDs forms a better MCQ prompt than the "cloze prompt", which is consistent with our observation. Besides, selecting answers by calculating and comparing the likelihoods of options is not as convenient and straightforward to implement as directly predicting option IDs. We thus suggest that *removing option IDs is not a practical method to mitigate selection bias*.

## 2.6 Can Simple Prompting Strategies Mitigate Selection Bias?

As a preliminary debiasing attempt, we apply two simple prompting strategies to `gpt-3.5-turbo`: (1) **Explicit debiasing instruction**: We append an explicit debiasing instruction in the system message of `gpt-3.5-turbo` ("Please note that the provided options have been randomly shuffled, so it is essential to consider them fairly and without bias."). (2) **Chain-of-Thought prompting** (Wei et al., 2022; Kojima et al., 2022): `gpt-3.5-turbo` is first prompted with "Let's think step by step:" to generate its thought process and then produces the final answer. We follow the implementation in OpenAI Evals, see Figure 10 in Appendix A for details. As shown in Table 2, the two prompting strategies cannot mitigate selection bias well. It suggests that *selection bias is an inherent behavioral bias of LLMs that cannot be addressed by simple prompt engineering*.

# 3 Methodology

## 3.1 Permutation-based Debiasing Baseline and Formulation

Before proposing our debiasing method, we first introduce a strong permutation-based debiasing baseline that our method builds upon. It averages the model's prediction distributions under various option permutations (Wang et al., 2023a; Zheng et al., 2023b), which intuitively cancels out both the model's token bias and position bias.

Formally, we use $q$ to denote the MCQ question. Suppose the $n$ default-ordered option IDs (e.g., A/B/C/D) are $d_i$ and the default-ordered option contents are $o_i$, $i \in \{1, 2, \ldots, n\}$. We use $I$ to denote a permutation of $\{1, 2, \ldots, n\}$, $\mathcal{I}$ to a set of possible $I$s. We use $g_I(i)$ to denote the index of $i$ in $I$, and $x^I$ to the concatenation of the default-ordered option IDs and the $I$-permuted option contents, so that $o_i$ is tied with $d_{g_I(i)}$ in $x^I$. The permutation-based debiasing baseline can be formulated as:

$$\widetilde{P}_{\text{debiased}}(o_i|q, x) = \frac{1}{|\mathcal{I}|} \sum_{I \in \mathcal{I}} P_{\text{observed}}(d_{g_I(i)}|q, x^I), \ \ i \in \{1, 2, \ldots, n\}, \tag{1}$$

where $x$ is the default input of option IDs and option contents, $P_{\text{observed}}(d_{g_I(i)}|q, x^I)$ is the observed prediction probability for the option ID $d_{g_I(i)}$ (meaning $o_i$ being correct) under the option permutation $I$, and $\widetilde{P}_{\text{debiased}}(o_i|q, x)$ denotes the debiased prediction probability for the option content $o_i$. Since computing full permutations is prohibitively expensive ($\times n!$ costs), we adopt a practical alternative, called **Cyclic Permutation**, where $\mathcal{I} = \{(i, i+1, \ldots, n, 1, \ldots, i-1)\}_{i=1}^{n}$. It reduces the computational cost (e.g., LLM forward times) from $\times n!$ to $\times n$ and ensures one pairing between each option ID $d_i$ and option content $o_j$. In Figure 16 in Appendix C, we show that selecting other permutations $\mathcal{I}$ in Cyclic Permutation, where we still ensure one pairing between each $d_i$ and $o_j$, leads to similar debiasing results. However, *the overhead of Cyclic Permutation is still somewhat high ($\times n$ inference costs), which stimulates us to design more efficient debiasing methods*.

## 3.2 Prediction Probability Decomposition

The core idea of our method PriDe is to obtain a debiased prediction distribution by *separating the model's prior bias for option IDs from the overall prediction distribution*. Equivalently, it assumes that the observed prediction distribution $P_{\text{observed}}$ over $d_i$ can be decomposed as a prior distribution $P_{\text{prior}}$ over $d_i$ and a debiased distribution $P_{\text{debiased}}$ over $o_i$:

$$P_{\text{observed}}(d_i|q, x^I) = Z_{q,x^I}^{-1} P_{\text{prior}}(d_i|q, x^I) P_{\text{debiased}}(o_{f_I(i)}|q, x^I), \quad \forall I \in \mathcal{I}, i \in \{1, 2, ..., n\}, \quad (2)$$

where $f_I(i)$ denotes $i$-th element in $I$. Note that we can rewrite the form of $P_{\text{observed}}$ as a joint probability $P(d_i, o_j|q, x^I)$ for $d_i$ and $o_j$, which equals to $P_{\text{observed}}(d_i|q, x^I)$ if $j = f_I(i)$ and 0 otherwise. Therefore, *the above prediction probability decomposition can be interpreted as the conditional independent assumption* (ignore the normalization item $Z_{q,x^I}$), where the model holds independent beliefs about $d_i$ and $o_j$. Specifically, $P_{\text{debiased}}(o_j|q, x^I)$ reflects the model's **true belief about the option content** $o_j$, which is not influenced by the option ID $d_i$. In contrast, $P_{\text{prior}}(d_i|q, x^I)$ indicates the model's **prior bias for the option ID** $d_i$, which actually involves not only the model's token bias but also position bias (§2.4), due to the natural binding between option IDs and options' ordering positions. Hence, under this formulation, we do not need to strictly distinguish the two biases laboriously, but can instead address them together by eliminating $P_{\text{prior}}$.

For tractable derivation, we reasonably assume that $P_{\text{debiased}}$ is not affected by how the options are ordered, which implies its invariance to option permutation $I$ so we can replace $x^I$ with the default input $x$. We also assume $P_{\text{prior}}$ to be independent of $x^I$, which means that the prior for option IDs depends on only the question $q$. Equation 2 is then simplified to:

$$P_{\text{observed}}(d_i|q, x^I) = Z_{q,x^I}^{-1} P_{\text{prior}}(d_i|q) P_{\text{debiased}}(o_{f_I(i)}|q, x), \quad \forall I \in \mathcal{I}, i \in \{1, 2, ..., n\}. \quad (3)$$

## 3.3 Debiasing with Prior Estimation

Taking the logarithm of both sides of Equation 3 and summing over all $I \in \mathcal{I}$, we can obtain:

$$\sum_{I \in \mathcal{I}} \log P_{\text{observed}}(d_i|q, x^I) = |\mathcal{I}| \log P_{\text{prior}}(d_i|q) + \left( \sum_{I \in \mathcal{I}} \log P_{\text{debiased}}(o_{f_I(i)}|q, x) \right) + C \quad (4)$$

$$= |\mathcal{I}| \log P_{\text{prior}}(d_i|q) + \left( \frac{|\mathcal{I}|}{n} \sum_{j=1}^{n} \log P_{\text{debiased}}(o_j|q, x) \right) + C \quad (5)$$

$$= |\mathcal{I}| \log P_{\text{prior}}(d_i|q) + C', \quad i \in \{1, 2, ..., n\}. \quad (6)$$

We derive Equation 5 because $\sum_{I \in \mathcal{I}} \log P_{\text{debiased}}(o_{f_I(i)}|q, x)$ actually involves $|\mathcal{I}|/n$ iterations over each $P_{\text{debiased}}(o_j|q, x)$ (given $\mathcal{I}$ contains either full or cyclic permutations), whose aggregation over $j \in \{1, 2, \ldots, n\}$ is a constant. Therefore, without any sample labels, we can obtain:

$$P_{\text{prior}}(d_i|q) = \text{softmax}\left( \frac{1}{|\mathcal{I}|} \sum_{I \in \mathcal{I}} \log P_{\text{observed}}(d_i|q, x^I) \right), \quad i \in \{1, 2, ..., n\}. \quad (7)$$

Recall our observation in §2.3 that selection bias within the same model displays a moderate cross-domain similarity. This implies that *the prior for option IDs is likely to transfer across different samples and domains*, which motivates us to compute the priors of partial test samples and use them as an approximation for the remaining samples. It can largely improve debiasing efficiency since no more computational overhead is needed for the remaining samples once the prior is estimated.

Drawing the above inspiration, PriDe first takes $K$ test samples $\mathcal{D}_e$ from the test set $\mathcal{D}$, where $K$ can be adjusted based on the estimation budget. Each sample in $\mathcal{D}_e$ undergoes the standard permutation-based debiasing in Equation 1, during which we estimate each sample-specific prior $P_{\text{prior}}(d_i|q)$ using Equation 7. For the remaining samples $\mathcal{D}_r = \mathcal{D} \backslash \mathcal{D}_e$, we compute the "global prior" $\widetilde{P}_{\text{prior}}(d_i)$ by averaging the previously computed priors as an approximation to the new sample's $P_{\text{prior}}(d_i|q)$ in Equation 3. We can thus compute the approximated $\widetilde{P}_{\text{debiased}}(o_i|q, x)$ and obtain the debiased prediction (omit the superscript $I$ as we only use the default input here):

$$\widetilde{P}_{\text{debiased}}(o_i|q, x) \propto P_{\text{observed}}(d_i|q, x) / \widetilde{P}_{\text{prior}}(d_i), \quad i \in \{1, 2, ..., n\}. \quad (8)$$

When $K \ll |\mathcal{D}|$, the overhead for prior estimation will become negligible compared to the whole inference cost. The overall procedure of PriDe is summarized as Algorithm 1.

---

**Algorithm 1** PriDe: Debiasing with Prior Estimation

---

**Require:** Language model, test samples $\mathcal{D} = \{(q_i, x_i)\}_i$, option number $n$, estimation budget $K$
**Ensure:** Model predictions $\mathcal{Y}$
 1: Initialize the model prediction set $\mathcal{Y} = \varnothing$ and the prior set $\mathcal{P} = \varnothing$     ▷ Initialization
 2: Sample the estimation samples $\mathcal{D}_e$ under $K$ and the remaining samples $\mathcal{D}_r = \mathcal{D} \backslash \mathcal{D}_e$
 3: **for** $(q, x) \in \mathcal{D}_e$ **do**
 4:     Debias the model prediction using Equation 1, add the predicted answer to $\mathcal{Y}$
 5:     Estimate the sample-specific prior $P_{\text{prior}}(d_i|q)$ using Equation 7, add it into $\mathcal{P}$
 6: **end for**
 7: Estimate the global prior $\widetilde{P}_{\text{prior}}(d_i)$ by averaging $\mathcal{P}$     ▷ Prior Estimation
 8: **for** $(q, x) \in \mathcal{D}_r$ **do**
 9:     Debias the model prediction using Equation 8     ▷ Efficient Debiasing
10:     Add the predicted answer to $\mathcal{Y}$
11: **end for**
12: **return** $\mathcal{Y}$

---

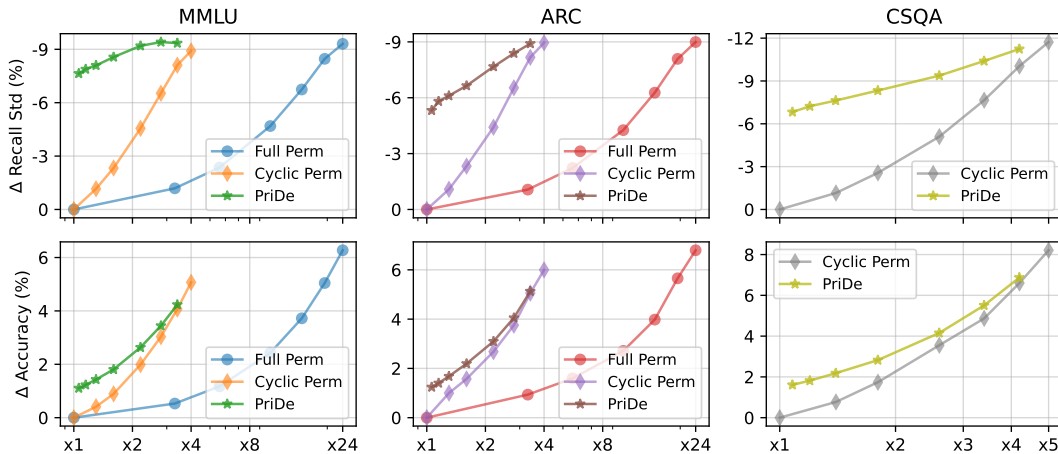

Figure 4: Debiasing results (0-shot, *averaged over all the 20 LLMs*) under varying computational costs (X-axis), see Table 3 and 4 in Appendix C for detailed breakdowns. We control the costs of Cyclic/Full Perm via the ratio $\beta$ of the debiased test samples, where we take $\beta \in \{0\%, 10\%, 20\%, 40\%, 60\%, 80\%, 100\%\}$. For PriDe, we control the costs via the ratio $\alpha$ of test samples for prior estimation (these samples are meanwhile directly debiased via Cyclic Perm), where we take $\alpha \in \{2\%, 5\%, 10\%, 20\%, 40\%, 60\%, 80\%\}$. Note that when $\alpha = 100\%$, PriDe degenerates to Cyclic Perm, so we do not plot them.

## 4 EXPERIMENTS

### 4.1 MAIN RESULTS

We compare PriDe with two strong permutation-based debiasing baselines: **Cyclic Permutation** and **Full Permutation**. The latter is not experimented on the 5-option CSQA benchmark due to the extremely high cost. Since `gpt-3.5-turbo` does not return the output probability, we sample 100 generated answers as an approximation to $P_{\text{observed}}$. For PriDe, we randomly sample $K = \alpha|\mathcal{D}|$ test samples as $\mathcal{D}_e$ and report the average results over 5 runs. Here, we use $\alpha \in (0, 1)$ as the ratio of test samples for prior estimation to control the estimation overhead.

Figure 4 presents the debiasing results (averaged over all the models) versus the computational costs under the 0-shot setting, see detailed breakdowns in Table 3 and 4 in Appendix C. *PriDe achieves superior debiasing effectiveness and performance improvement to Full/Cyclic Perm, especially in the low-cost scenario.* This also holds under the 5-shot setting, see Figure 19 in Appendix C. In Figure 17 in Appendix C, we show that the estimated prior manifests a clear correlation with the empirical selection bias (i.e., the recalls of different option positions before debiasing). It suggests that *PriDe can provide a good interpretation for the model's selection bias.* Furthermore, we observe that the priors are stable when estimated with different sizes of test samples (from 2% to 20%). It suggests that *we are able to obtain a reliable estimate of prior even with a limited computational*

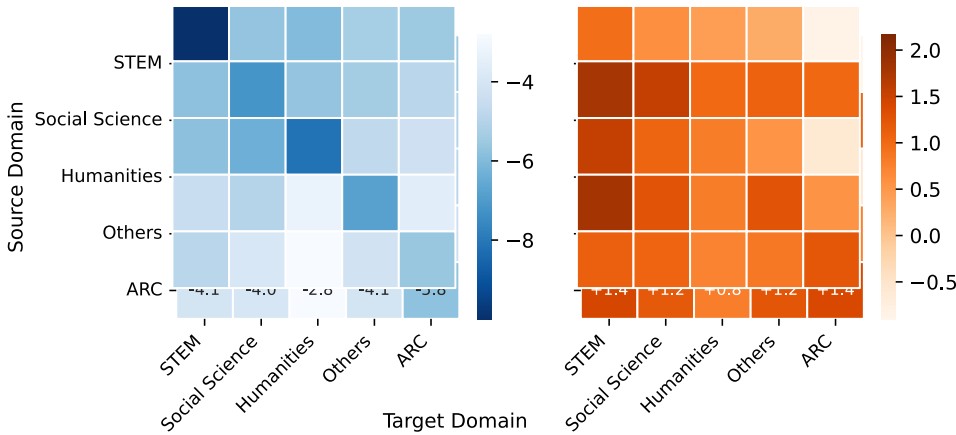

Figure 5: Cross-domain debiasing results of PriDe (0-shot, *averaged over all the 20 LLMs*), with priors estimated using $\alpha = 5\%$ test samples from the source domains.

*budget*. It also implies that the model's prior bias for option IDs exhibits a similar pattern across different samples, confirming our design motivation of PriDe in §3.3.

Note that one may propose to debias with fewer permutations (such as 2 or 3 random permutations, with $\times 2$ or $\times 3$ costs) as a low-cost alternative to Cyclic ($\times n$) or Full ($\times n!$) Perm. In Figure 21 and 22 in Appendix C, we show that PriDe can still be combined with these methods and notably boost debiasing effectiveness and efficiency.

## 4.2 TRANSFERABILITY ANALYSIS

In practical scenarios, we may not guarantee that the test samples always come from the same or similar domains. We hope the prior estimated by PriDe to be transferable: Once the prior is estimated using a small number of samples from domain X, it can be used to debias not only other samples from domain X but also samples from domain Y. To this end, we evaluate the cross-domain transfer performance of estimated priors on the four category domains of MMLU and ARC (5 domains in total, all are 4-option MCQs). We first use PriDe to estimate the prior with $\alpha$ test samples from a source domain X, and then apply it to debiasing the test samples from a target domain Y. As shown in Figure 5, *the estimated priors exhibit reasonable transferability across different domains*.

However, we find that although the transferability in terms of debiasing is promising, there may be a slight degradation in model performance when the domain gap is large (e.g., from STEM/Humanities to ARC). While PriDe is not designed to improve model performance but rather to mitigate selection bias and thereby improve LLMs' robustness, we suggest *updating the estimated prior using new samples when there are predictable domain shifts in test samples*, whose overhead is still negligible compared to the whole inference cost.

## 4.3 HOW DEBIASING AFFECTS MODEL PREDICTIONS?

We notice that although it is not our initial intention, the debiasing methods (PriDe and Cyclic/Full Perm) usually improve the model performance. With PriDe ($\alpha = 5\%$) and Cyclic Perm as examples, we seek further insights on how debiasing affects model predictions and consequently improves model performance. We are especially interested in how PriDe works with the estimated prior and how it works differently from Cyclic Perm.

We put the breakdown of the model prediction changes in Table 5 and 6 in Appendix C, and summarize our main observations as follows: (1) For both PriDe and Cyclic Perm, the samples with predictions changing from wrong to correct always surpass those from correct to wrong, which illustrates the improvement in model performance. (2) For the samples where the predictions are altered after PriDe debiasing, the model usually holds low confidence in the original predictions. Meanwhile, the debiased predictions typically rank top-2 in the original prediction distribution, which holds especially for the larger models. It is because the predictions that the model is uncertain about are more susceptible to the model's prior bias for option IDs, which is exactly what PriDe aims to al-

leviate. (3) Cyclic Perm even alters high-confidence predictions and brings more drastic prediction changes, e.g., the lowest-ranked option becomes the debiased prediction more frequently than in PriDe. It also remarkably flattens the prediction distribution, i.e., $\widetilde{P}_{\text{debiased}}(o_i|q, x)$ is smoother than $P_{\text{observed}}(d_i|q, x)$. This is probably because the permutation-based debiasing is essentially a kind of "mixture of experts", where each "expert model" prepends the LLM with an option permutation operation. It leads to more deliberate and moderately confident predictions (corresponding to a flatter distribution), but on the other hand, requires more computational overhead than PriDe.

## 5 RELATED WORK

**Large Language Models (LLMs)**  The realm of large language models (LLMs) has been undergoing rapid and significant developments since the launch of GPT-3 (Brown et al., 2020). Prominent examples, such as ChatGPT (OpenAI, 2022), LLaMA (Touvron et al., 2023a;b), Alpaca (Taori et al., 2023), and Vicuna (Chiang et al., 2023), have emerged successively over the past year. These models usually boast billions of parameters and are trained to comprehend natural language and follow human instructions (Ouyang et al., 2022; Wang et al., 2023b).

**Multiple Choice Questions (MCQs)**  As a concise task format, MCQs are widely adopted in LLM-centric scenarios. For instance, in the automatic evaluation framework of Chiang et al. (2023); Zheng et al. (2023b), GPT-4 (OpenAI, 2023) is presented with a question and two model answers and is tasked to determine which one is better. Numerous standard language model benchmarks also employ the task format of MCQs, such as MMLU (Hendrycks et al., 2020), ARC (Hendrycks et al., 2020), AGIEval (Zhong et al., 2023), C-Eval (Huang et al., 2023). It thus piques our research interest in the robustness of LLMs in the context of MCQs.

**Bias and Robustness of LLMs**  In our study, we use "bias" to refer to the systematic error within LLMs rather than the prejudice in culture or gender studied in the field of LLM safety (Zheng et al., 2023a; 2024). The bias in language models has always been an important research area closely related to model robustness. For instance, Zhao et al. (2021) showed that GPT-3 is sensitive to task instructions and in-context examples, which arises from its bias towards certain answers. Chen et al. (2022); Si et al. (2023); Pan et al. (2023) explored how LLMs' few-shot learning is influenced by the construction of in-context examples. Wang et al. (2023a); Zheng et al. (2023b) found that GPT-4 leans towards the first presented answers and may produce unfair evaluation results.

Contemporaneous with our work, Pezeshkpour & Hruschka (2023) similarly observed that LLMs are sensitive to option position changes in MCQs and verified position bias as one cause of sensitivity. However, their study does not ablate the impact of option IDs, making their investigation on "position bias" less convincing. In fact, our study finds that position bias is less regular and largely depends on models and tasks, while token bias is a more salient intrinsic cause of LLMs' selection bias and, consequently, poor robustness. Their study is also conducted on very limited models (only GPT-4 and InstructGPT without open-source models, somewhat hindering reproducibility) and tasks (only three MMLU subtasks, one Big-Bench subtask (Srivastava et al., 2023), and CSQA). In contrast, we draw more general observations through extensive cross-model and cross-task empirical evaluation, which further inspires our proposal of the computation-efficient, interpretable, and transferable debiasing method PriDe.

## 6 CONCLUSION

This work studies the inherent selection bias of large language models (LLMs), which makes them vulnerable to option position changes in multiple choice question (MCQ) evaluation. Through extensive empirical analyses, we pinpoint that this behavioral bias stems primarily from token bias, where the model a priori assigns more probabilistic mass to specific option ID tokens when predicting answers from the option IDs, and partially from position bias, where the model favors options presented at specific ordering positions. In particular, token bias is a more salient intrinsic cause of selection bias, while position bias is less regular and depends on models and tasks. Our proposed debiasing method PriDe estimates the model's prior bias for option IDs and separates it from the overall prediction distribution. It remarkably mitigates selection bias, with no need for sample labels and only negligible computational overhead. We especially highlight PriDe's high efficiency, interpretability, and cross-domain transferability. We hope the empirical analyses in this work and our debiasing method can inspire future research on the bias and robustness of LLMs.

## REPRODUCIBILITY STATEMENT

We have released the evaluation data, code, and experimental results at `https://github.com/chujiezheng/LLM-MCQ-Bias` to facilitate reproducible research. Our experiments were run on A100 40GB GPUs (for 70B models) and V100 32GB (for other models). The open-source models and data used in this work are listed in Appendix G and Appendix E, respectively.

## ACKNOWLEDGMENTS

This work was supported by the National Science Foundation for Distinguished Young Scholars (with No. 62125604). This work was also supported by the NSFC projects (Key project with No. 61936010).

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

# A PROMPTS USED IN EXPERIMENTS

```
The following are multiple choice questions about electrical engineering.  You
should directly answer the question by choosing the correct option.

[ in-context examples (if few-shot) ]

Question:  In an SR latch built from NOR gates, which condition is not allowed
Options:
A. S=0, R=0
B. S=0, R=1
C. S=1, R=0
D. S=1, R=1
Answer: D
```

Figure 6: Input formats for open-source models (e.g., `llama(-2)` and `vicuna`). The `black text` is the templated input. The next-token prediction probabilities of the option IDs at the red text is used as the observed prediction distribution. Note that for all the open-source models, we do not prepend the input text with a `bos_token`.

```
{ "role":  "system", "content":  "The following are multiple choice questions about
electrical engineering.  You should directly answer the question by choosing the
correct option." }

[ in-context examples (if few-shot) ]

{ "role":  "user", "content":  """Question:  In an SR latch built from NOR gates,
which condition is not allowed
Options:
A. S=0, R=0
B. S=0, R=1
C. S=1, R=0
D. S=1, R=1
Answer:""" }

{ "role":  "assistant", "content":  "D" }
```

Figure 7: Input formats for `gpt-3.5-turbo`. The `black text` is the templated API request. The blue text is in-context or test samples. The red text is the expected API-returned result. The orange text is the subject in MMLU (57 different ones in total).

```
The following are multiple choice questions about electrical engineering.  You
should directly answer the question by choosing the correct option.

[ in-context examples (if few-shot) ]

Question:  In an SR latch built from NOR gates, which condition is not allowed
Options:
S=0, R=0
S=0, R=1
S=1, R=0
S=1, R=1
Answer: S=1, R=1
```

Figure 8: Input formats for open-source models with option IDs removed. We compute the likelihoods of options, normalized by their lengths, and use the maximum one as the model prediction.

```
{ "role":  "system", "content":  "The following are multiple choice questions about
electrical engineering.  You should directly answer the question by choosing the
correct option." }

[ in-context examples (if few-shot) ]

{ "role":  "user", "content":  """Question:  In an SR latch built from NOR gates,
which condition is not allowed
Options:
S=0, R=0
S=0, R=1
S=1, R=0
S=1, R=1
Answer:""" }

{ "role":  "assistant", "content":  "S=1, R=1" }
```

Figure 9: Input formats for `gpt-3.5-turbo` with option IDs removed. We require it to generate the whole selected option, which is then compared with the golden answer.

The first API call

```
{ "role":  "system", "content":  "The following are multiple choice questions about
electrical engineering.  You should reason in a step-by-step manner as to get the
right answer." }
{ "role":  "user", "content":  """Question:  In an SR latch built from NOR gates,
which condition is not allowed
Options:
A. S=0, R=0
B. S=0, R=1
C. S=1, R=0
D. S=1, R=1
Answer:  Let's think step by step:""" }

{ "role":  "assistant", "content":  "[ thought or reasoning process ]" }
```

The second API call

```
{ "role":  "system", "content":  "The following are multiple choice questions about
electrical engineering.  You should reason in a step-by-step manner as to get the
right answer." }
{ "role":  "user", "content":  """Question:  In an SR latch built from NOR gates,
which condition is not allowed
Options:
A. S=0, R=0
B. S=0, R=1
C. S=1, R=0
D. S=1, R=1
Answer:  Let's think step by step:""" }
{ "role":  "assistant", "content":  "[ thought or reasoning process ]" }
{ "role":  "assistant", "content":  "Given all of the above, the answer of the
question is:" }

{ "role":  "assistant", "content":  "D" }
```

Figure 10: Chain-of-Thought prompt for `gpt-3.5-turbo`, generally following the implementation of OpenAI Evals, where the decoding temperature is set to 0.

# B    EVALUATION RESULTS OF SELECTION BIAS

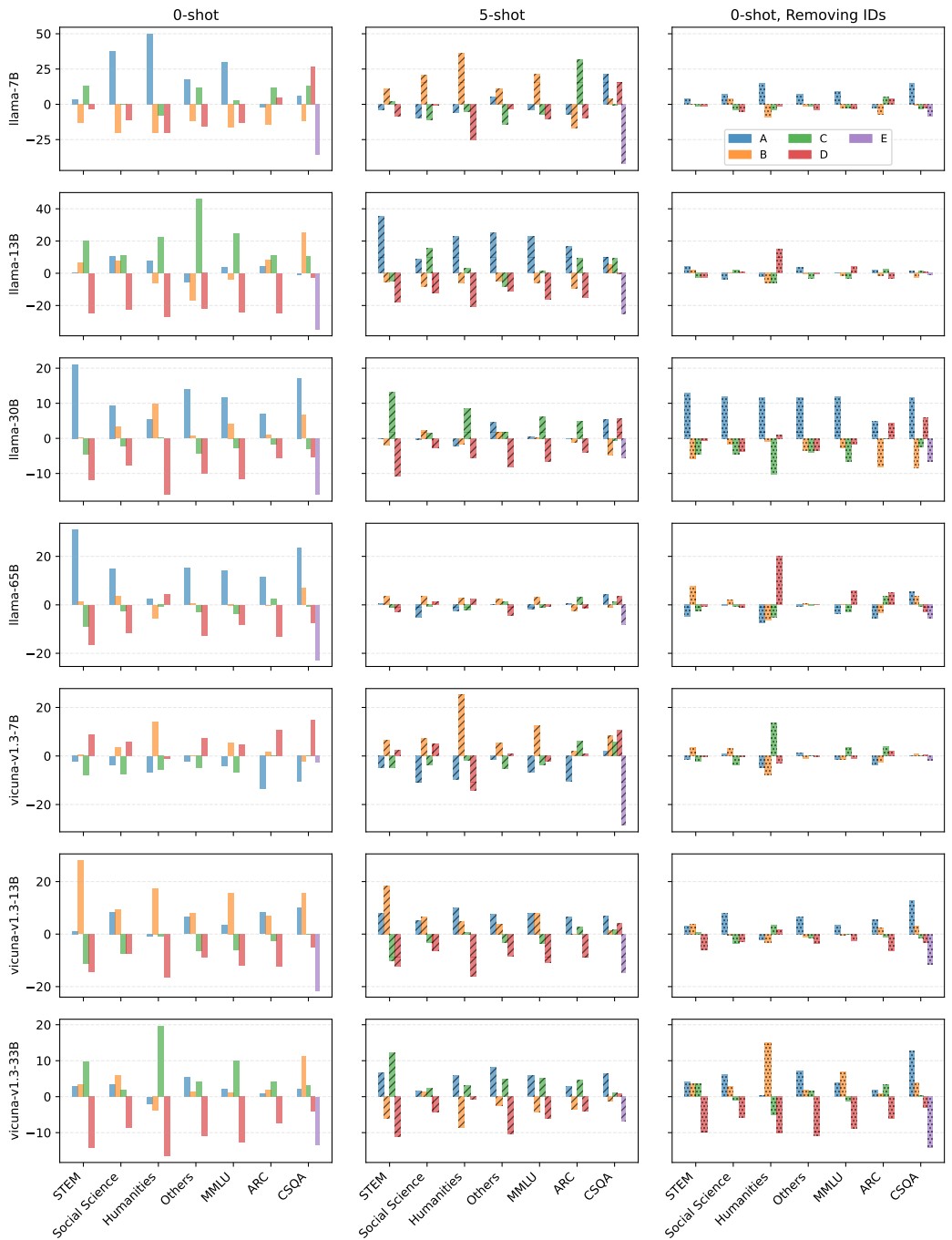

Figure 11: Selection bias of `llama` and `vicuna-v1.3` models. Y-axis: the recall score (%) normalized by *subtracting the overall accuracy*.

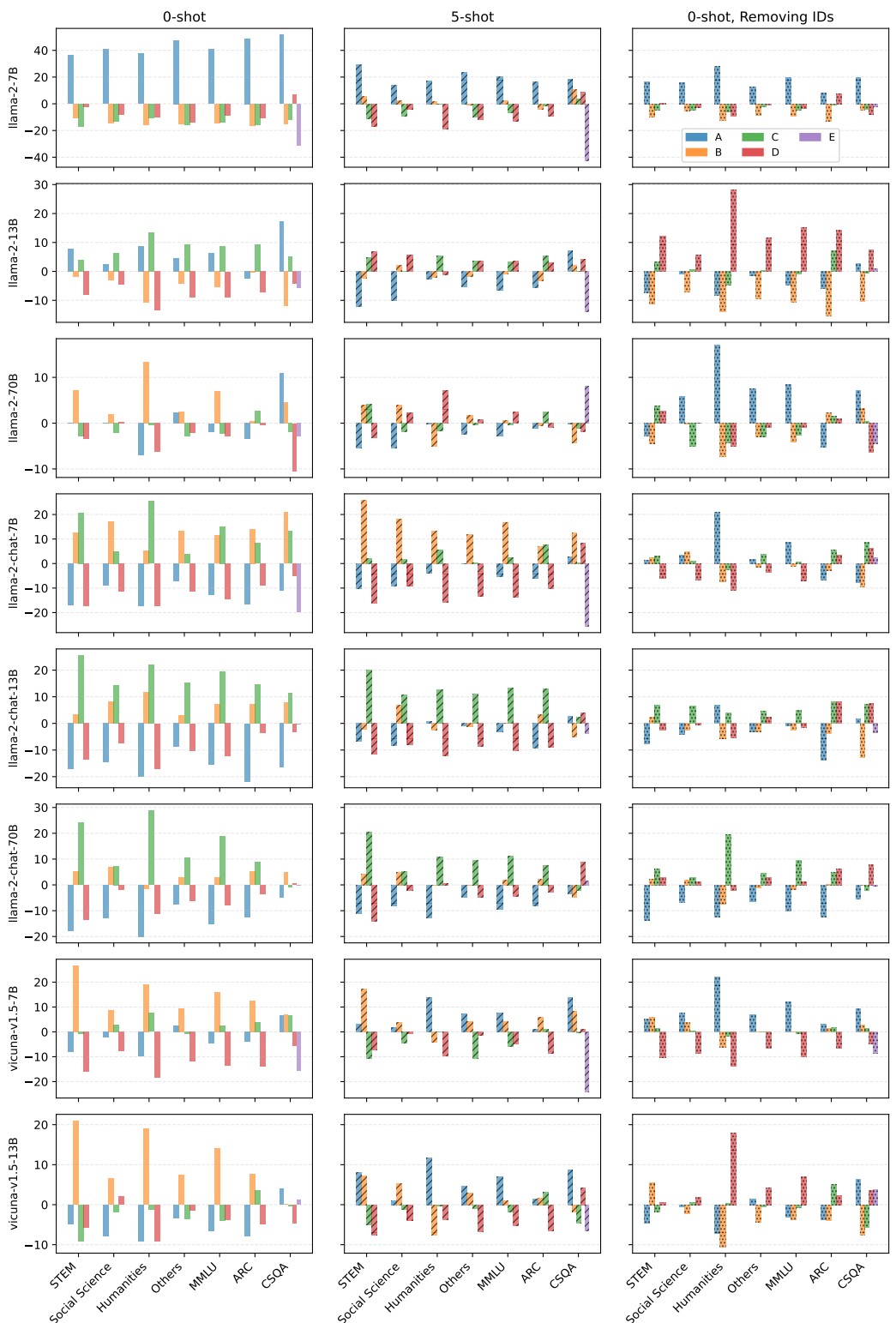

Figure 12: Selection bias of `llama-2`, `llama-2-chat`, and `vicuna-v1.5` models.

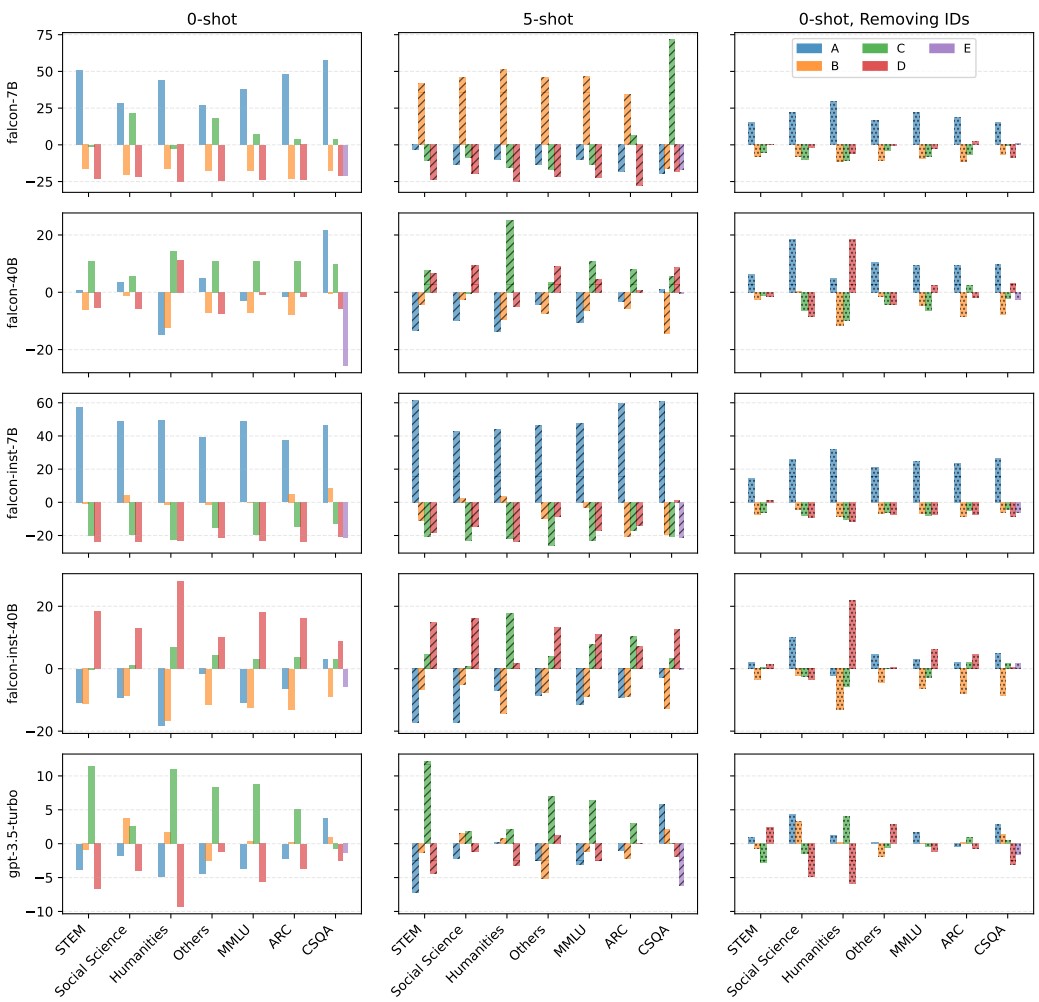

Figure 13: Selection bias of `falcon` and `gpt-3.5-turbo` models. Note that we conjecture `falcon-7B` and `falcon-inst-7B` are undertrained. This is evidenced by: (1) they exhibits abnormally pronounced selection bias in various benchmarks, as reflected in their Y-axis scales, (2) their performance on MMLU is almost random selection (even 5-shot, see Table 3 and Table 4 in Appendix C), and (3) their recalls of position D are close to zero (see Figure 17 in Appendix C).

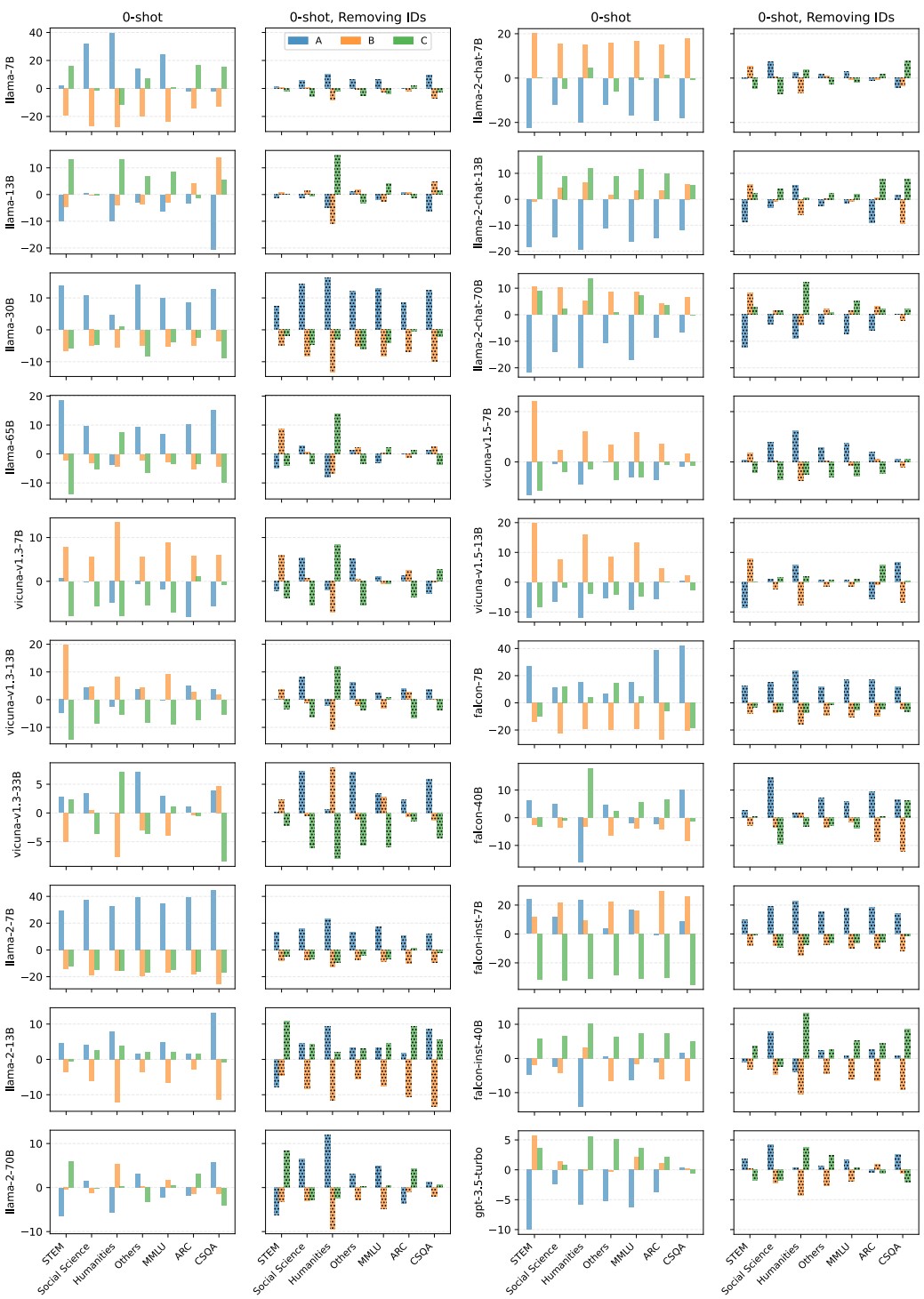

Figure 14: Selection bias evaluated under the 3-option setting, where we randomly sampled three options (including the golden answer) from the original candidate options. Removing option IDs notably reduces selection bias for most LLMs, except `llama-30B`, `vicuna-v1.3-33B`, and `llama-2-13/70B`.

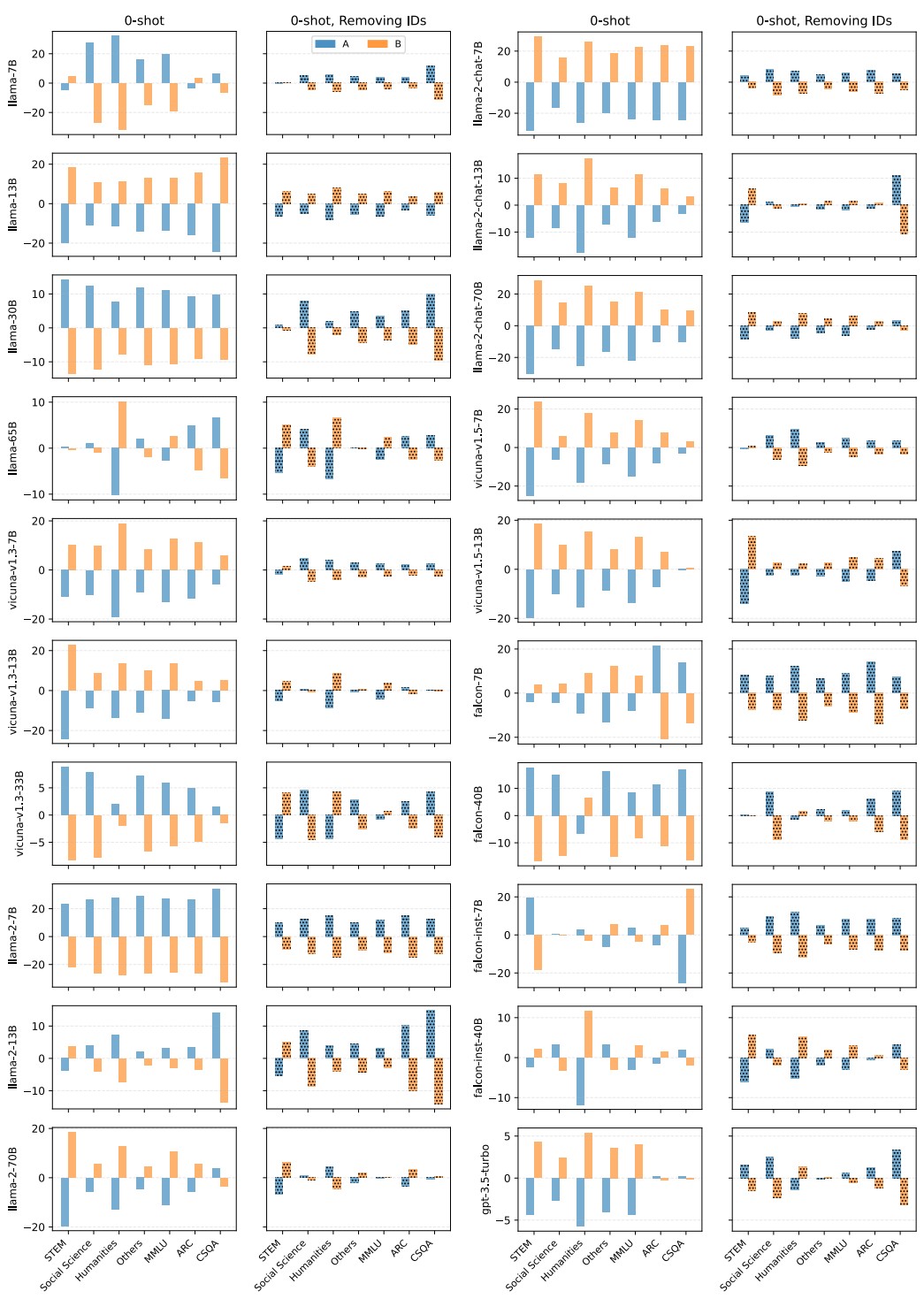

Figure 15: Selection bias evaluated under the 2-option setting, where we randomly sampled two options (including the golden answer) from the original candidate options. Removing option IDs notably reduces selection bias for most LLMs, except `llama-2-13B` and `falcon(-inst)-7B` (the latter is likely to be undertrained).

# C    SUPPLEMENTARY EXPERIMENTAL RESULTS

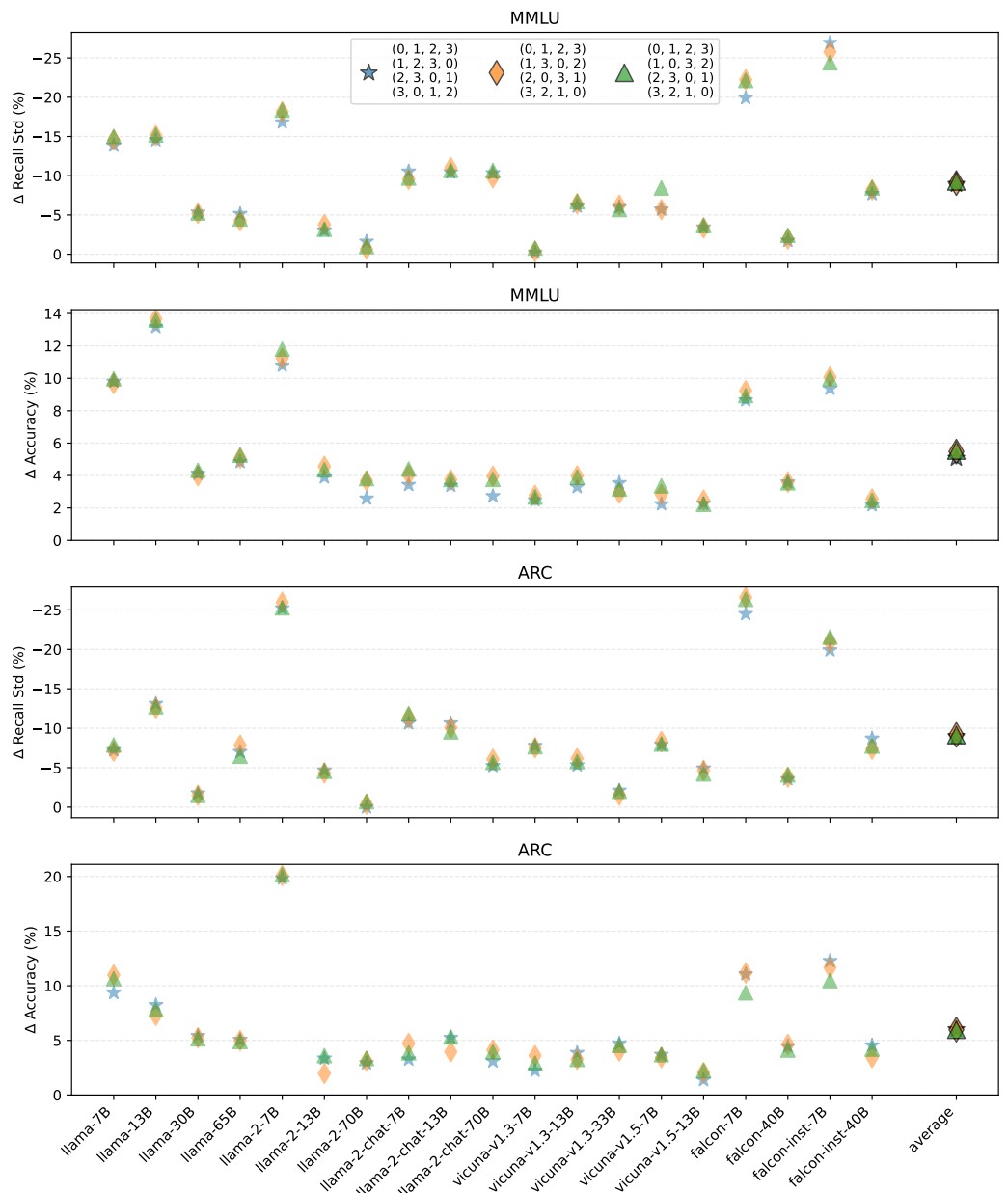

Figure 16: Debiasing results (0-shot, MMLU and ARC) of Cyclic Permutation with different selected permutations $\mathcal{I}$. The selection of permutations ensures one pairing between each option ID $d_i$ and option content $o_j$. We do not observe remarkable differences between these selections.

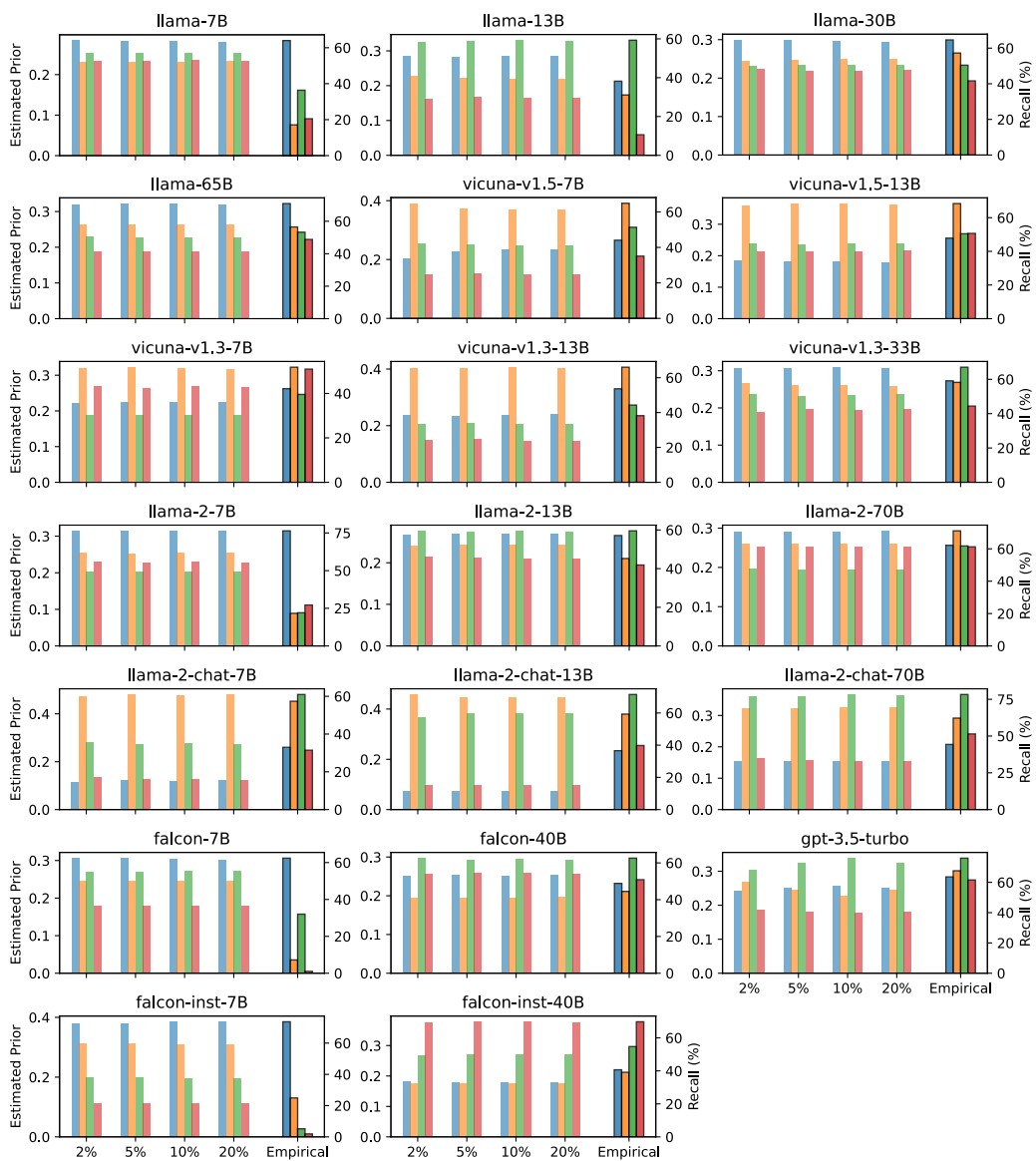

Figure 17: Prior preferences estimated by PriDe with varying sizes of test samples (2%, 5%, 10%, 20%) versus the empirical selection bias of the original model predictions (0-shot MMLU).

We observe that the estimated priors usually correlate strongly with the empirical selection bias, except in a few cases where the correlation is moderate, like in `llama-2-7B`, `llama-2-chat-7B`, and `vicuna-v1.3-33B`. Specifically, the estimated prior probabilities may not necessarily align perfectly with the frequency-based recalls. Even in the cases where the rankings are the same, the scales may still differ, such as in `llama-7B/13B`. Our preliminary investigation suggests that it may be related to the calibration of LLMs (Kadavath et al., 2022). We conjecture that an ideal alignment requires the model to be well-calibrated, i.e., the probabilistic predictions match up with frequencies of correctness (Jiang et al., 2021). We leave this research question for future work.

Table 3: Breakdown of the debiasing results (0-shot), measured by standard deviation of recalls (RStd) and accuracy (Acc).

| Models \ Methods | Default RStd | Acc | Removing IDs RStd | Acc | Cyclic Perm RStd | Acc | PriDe (5%) RStd | Acc | PriDe (40%) RStd | Acc | PriDe (80%) RStd | Acc |
|---|---|---|---|---|---|---|---|---|---|---|---|---|
| MMLU | Cost: ×1 | | Cost: ×1 | | Cost: ×4 | | Cost: ×1.15 | | Cost: ×2.2 | | Cost: ×3.4 | |
| llama-7B | 18.5 | 33.7 | 5.2 | 33.4 | 4.7 | 43.5 | 5.5 | 35.3 | 5.1 | 38.4 | 4.9 | 41.8 |
| llama-13B | 17.4 | 34.6 | 2.6 | 29.1 | 2.9 | 47.7 | 5.7 | 36.4 | 3.9 | 40.4 | 2.6 | 45.3 |
| llama-30B | 8.5 | 53.1 | 7.0 | 49.3 | 3.2 | 57.2 | 3.6 | 54.0 | 3.4 | 55.1 | 3.1 | 56.4 |
| llama-65B | 8.3 | 56.9 | 3.6 | 51.4 | 3.1 | 61.8 | 1.9 | 58.5 | 1.2 | 59.8 | 2.3 | 61.1 |
| llama-2-7B | 23.0 | 35.8 | 11.2 | 37.0 | 6.1 | 46.6 | 5.5 | 40.2 | 4.9 | 42.6 | 5.5 | 45.3 |
| llama-2-13B | 7.5 | 50.8 | 9.6 | 44.7 | 4.5 | 54.7 | 2.3 | 51.8 | 2.2 | 52.9 | 3.5 | 54.0 |
| llama-2-70B | 4.1 | 64.0 | 4.9 | 58.9 | 2.5 | 66.6 | 5.7 | 64.3 | 3.5 | 65.2 | 2.1 | 66.2 |
| llama-2-chat-7B | 13.5 | 45.8 | 5.7 | 45.0 | 3.0 | 49.3 | 7.9 | 46.4 | 5.4 | 47.5 | 3.2 | 48.7 |
| llama-2-chat-13B | 14.3 | 52.1 | 2.9 | 51.0 | 3.9 | 55.4 | 8.2 | 53.5 | 4.7 | 54.2 | 2.9 | 55.0 |
| llama-2-chat-70B | 12.8 | 59.4 | 7.0 | 53.4 | 2.5 | 62.1 | 9.0 | 60.2 | 6.1 | 60.9 | 3.3 | 61.7 |
| vicuna-v1.3-7B | 5.3 | 46.2 | 1.9 | 44.2 | 5.1 | 48.7 | 3.7 | 46.6 | 3.1 | 47.4 | 4.3 | 48.4 |
| vicuna-v1.3-13B | 10.5 | 50.2 | 2.0 | 49.8 | 4.4 | 53.5 | 3.6 | 50.8 | 3.5 | 51.8 | 4.1 | 53.0 |
| vicuna-v1.3-33B | 8.2 | 57.0 | 6.0 | 57.1 | 2.2 | 60.6 | 7.0 | 57.6 | 5.0 | 58.6 | 2.9 | 59.9 |
| vicuna-v1.5-7B | 10.9 | 48.7 | 7.9 | 46.5 | 5.2 | 51.0 | 5.2 | 49.1 | 3.3 | 49.8 | 4.0 | 50.5 |
| vicuna-v1.5-13B | 8.2 | 54.4 | 4.3 | 53.1 | 4.8 | 56.6 | 2.5 | 55.2 | 2.2 | 55.8 | 3.7 | 56.2 |
| falcon-7B | 24.2 | 24.7 | 12.6 | 30.7 | 4.3 | 33.3 | 4.2 | 27.0 | 3.4 | 29.2 | 3.8 | 31.9 |
| falcon-40B | 6.7 | 51.8 | 6.1 | 44.2 | 5.0 | 55.4 | 2.6 | 52.3 | 1.6 | 53.4 | 3.5 | 54.6 |
| falcon-inst-7B | 28.7 | 25.0 | 13.7 | 29.9 | 1.7 | 34.4 | 7.3 | 28.1 | 4.8 | 30.4 | 2.2 | 33.0 |
| falcon-inst-40B | 12.4 | 51.5 | 4.8 | 45.1 | 4.7 | 53.7 | 1.7 | 51.8 | 1.0 | 52.5 | 3.3 | 53.3 |
| gpt-3.5-turbo | 5.5 | 67.2 | 1.0 | 66.7 | 1.1 | 69.8 | 2.8 | 68.0 | 2.2 | 68.5 | 1.2 | 69.3 |
| Δ Average | | | -6.4 | -2.1 | -8.7 | +4.9 | -7.6 | +1.2 | -8.9 | +2.6 | -9.1 | +4.1 |
| ARC | Cost: ×1 | | Cost: ×1 | | Cost: ×4 | | Cost: ×1.15 | | Cost: ×2.2 | | Cost: ×3.4 | |
| llama-7B | 9.9 | 38.6 | 4.9 | 36.0 | 2.7 | 48.0 | 4.0 | 40.1 | 2.3 | 43.3 | 2.6 | 46.4 |
| llama-13B | 14.2 | 51.3 | 2.3 | 27.8 | 1.1 | 59.6 | 7.7 | 51.6 | 4.2 | 55.2 | 1.9 | 58.3 |
| llama-30B | 4.6 | 67.6 | 5.3 | 63.5 | 2.8 | 73.0 | 3.7 | 67.1 | 2.2 | 68.7 | 2.0 | 71.6 |
| llama-65B | 8.8 | 72.8 | 4.4 | 67.6 | 1.8 | 77.9 | 1.9 | 73.0 | 1.7 | 74.6 | 1.6 | 76.9 |
| llama-2-7B | 27.4 | 36.0 | 8.5 | 40.7 | 2.1 | 55.8 | 7.8 | 45.8 | 4.5 | 49.4 | 2.1 | 53.7 |
| llama-2-13B | 6.0 | 62.9 | 11.5 | 53.0 | 1.4 | 66.3 | 4.8 | 63.9 | 3.1 | 64.9 | 2.1 | 66.0 |
| llama-2-70B | 2.3 | 80.7 | 3.0 | 78.7 | 2.3 | 83.6 | 5.2 | 80.9 | 3.6 | 82.0 | 2.3 | 83.1 |
| llama-2-chat-7B | 12.4 | 56.5 | 4.8 | 53.7 | 1.8 | 59.7 | 7.3 | 57.5 | 4.7 | 58.4 | 2.6 | 59.9 |
| llama-2-chat-13B | 13.7 | 64.4 | 9.2 | 64.0 | 3.0 | 69.6 | 8.5 | 66.1 | 5.6 | 67.2 | 3.2 | 68.9 |
| llama-2-chat-70B | 8.2 | 78.0 | 7.3 | 71.7 | 3.0 | 81.1 | 7.4 | 78.2 | 5.6 | 79.3 | 3.5 | 80.6 |
| vicuna-v1.3-7B | 8.6 | 53.5 | 3.1 | 53.6 | 0.9 | 55.7 | 4.2 | 53.8 | 2.7 | 54.7 | 1.1 | 55.1 |
| vicuna-v1.3-13B | 8.3 | 62.9 | 4.3 | 61.5 | 2.9 | 66.8 | 4.7 | 64.1 | 2.0 | 64.6 | 1.7 | 66.2 |
| vicuna-v1.3-33B | 4.4 | 72.9 | 3.6 | 74.3 | 2.2 | 77.6 | 4.1 | 72.9 | 2.8 | 74.8 | 2.1 | 76.6 |
| vicuna-v1.5-7B | 9.8 | 58.5 | 3.8 | 56.5 | 1.9 | 62.1 | 5.8 | 59.0 | 4.2 | 60.3 | 2.4 | 61.5 |
| vicuna-v1.5-13B | 6.2 | 69.7 | 3.9 | 67.7 | 1.3 | 71.1 | 3.7 | 69.8 | 2.8 | 70.2 | 1.8 | 71.0 |
| falcon-7B | 29.2 | 24.9 | 11.5 | 29.5 | 4.7 | 36.0 | 3.9 | 27.8 | 3.7 | 30.9 | 4.0 | 34.4 |
| falcon-40B | 6.8 | 62.9 | 6.4 | 55.7 | 3.3 | 67.4 | 5.6 | 64.1 | 4.0 | 65.4 | 3.2 | 66.7 |
| falcon-inst-7B | 23.4 | 25.2 | 13.0 | 29.7 | 3.5 | 37.4 | 10.2 | 27.8 | 6.5 | 31.2 | 3.2 | 35.8 |
| falcon-inst-40B | 11.1 | 61.3 | 4.8 | 56.1 | 2.4 | 65.8 | 4.3 | 63.6 | 3.4 | 64.3 | 2.7 | 65.3 |
| gpt-3.5-turbo | 3.3 | 84.3 | 0.6 | 84.9 | 1.6 | 85.0 | 2.3 | 84.2 | 2.1 | 84.1 | 1.7 | 84.9 |
| Δ Average | | | -5.1 | -2.9 | -8.6 | +5.7 | -5.6 | +1.3 | -7.3 | +2.9 | -8.5 | +4.9 |
| CSQA | Cost: ×1 | | Cost: ×1 | | Cost: ×5 | | Cost: ×1.2 | | Cost: ×2.6 | | Cost: ×4.2 | |
| llama-7B | 21.6 | 36.4 | 7.9 | 26.6 | 2.9 | 53.0 | 8.5 | 38.5 | 4.9 | 45.2 | 3.3 | 50.5 |
| llama-13B | 19.9 | 55.3 | 1.6 | 39.4 | 2.3 | 64.2 | 9.5 | 59.4 | 6.7 | 60.6 | 3.7 | 63.1 |
| llama-30B | 11.2 | 66.0 | 7.6 | 52.6 | 2.7 | 71.1 | 9.0 | 65.8 | 6.8 | 68.2 | 3.9 | 70.0 |
| llama-65B | 15.3 | 65.0 | 4.0 | 56.9 | 1.3 | 74.4 | 4.8 | 68.1 | 3.7 | 70.4 | 2.3 | 73.3 |
| llama-2-7B | 28.4 | 31.9 | 9.9 | 32.2 | 0.8 | 56.0 | 9.9 | 41.5 | 6.0 | 46.3 | 2.6 | 52.9 |
| llama-2-13B | 10.2 | 57.0 | 5.8 | 43.8 | 1.3 | 66.6 | 7.4 | 58.4 | 5.7 | 61.4 | 2.3 | 64.8 |
| llama-2-70B | 7.3 | 70.9 | 4.9 | 65.5 | 3.0 | 75.5 | 6.0 | 71.6 | 4.7 | 73.0 | 3.6 | 74.8 |
| llama-2-chat-7B | 15.2 | 56.5 | 7.4 | 53.5 | 1.2 | 64.8 | 10.1 | 59.5 | 6.3 | 61.6 | 2.4 | 63.4 |
| llama-2-chat-13B | 9.8 | 64.0 | 7.5 | 54.2 | 2.0 | 68.8 | 6.3 | 65.2 | 3.6 | 66.4 | 1.3 | 68.0 |
| llama-2-chat-70B | 3.1 | 74.6 | 4.4 | 56.2 | 0.7 | 76.6 | 3.1 | 74.8 | 1.7 | 75.5 | 1.1 | 76.4 |
| vicuna-v1.3-7B | 8.3 | 56.9 | 0.9 | 57.2 | 1.8 | 62.8 | 4.0 | 57.9 | 2.4 | 59.4 | 1.4 | 61.5 |
| vicuna-v1.3-13B | 12.9 | 63.4 | 8.0 | 63.2 | 3.8 | 69.9 | 6.6 | 65.3 | 5.3 | 66.9 | 4.0 | 68.9 |
| vicuna-v1.3-33B | 8.2 | 68.9 | 8.8 | 70.1 | 2.7 | 72.8 | 5.1 | 69.9 | 3.9 | 70.9 | 2.7 | 72.0 |
| vicuna-v1.5-7B | 9.1 | 60.4 | 6.2 | 56.4 | 0.8 | 64.7 | 4.1 | 61.1 | 2.9 | 62.3 | 1.5 | 64.2 |
| vicuna-v1.5-13B | 2.8 | 67.1 | 5.6 | 62.1 | 1.3 | 70.3 | 2.5 | 67.1 | 2.1 | 68.4 | 1.4 | 69.6 |
| falcon-7B | 29.9 | 20.9 | 8.2 | 30.8 | 3.3 | 28.9 | 6.2 | 20.4 | 3.7 | 23.5 | 2.9 | 27.2 |
| falcon-40B | 15.9 | 61.4 | 6.0 | 54.1 | 1.8 | 70.4 | 9.2 | 63.2 | 5.8 | 65.8 | 2.8 | 69.0 |
| falcon-inst-7B | 25.7 | 21.7 | 13.1 | 34.8 | 3.3 | 39.4 | 8.7 | 25.5 | 4.9 | 29.9 | 3.2 | 36.1 |
| falcon-inst-40B | 6.5 | 67.2 | 4.5 | 54.3 | 1.4 | 71.4 | 3.1 | 66.9 | 1.9 | 68.7 | 1.6 | 70.6 |
| gpt-3.5-turbo | 2.2 | 76.1 | 2.1 | 76.1 | 1.2 | 78.0 | 2.2 | 76.4 | 1.9 | 77.0 | 1.6 | 77.8 |
| Δ Average | | | -6.2 | -7.0 | -11.2 | +7.9 | -6.9 | +1.7 | -8.9 | +4.0 | -10.7 | +6.6 |

Table 4: Breakdown of the debiasing results (5-shot).

| Models \ Methods | Default | | Removing IDs | | Cyclic Perm | | PriDe (5%) | | PriDe (40%) | | PriDe (80%) | |
|---|---|---|---|---|---|---|---|---|---|---|---|---|
| | RStd | Acc | RStd | Acc | RStd | Acc | RStd | Acc | RStd | Acc | RStd | Acc |
| MMLU | Cost: ×1 | | Cost: ×1 | | Cost: ×4 | | Cost: ×1.15 | | Cost: ×2.2 | | Cost: ×3.4 | |
| llama-7B | 12.5 | 34.5 | 2.8 | 39.3 | 4.8 | 44.5 | 2.6 | 35.4 | 2.4 | 38.6 | 4.2 | 42.4 |
| llama-13B | 14.3 | 43.5 | 2.6 | 45.0 | 4.7 | 51.1 | 4.6 | 45.7 | 4.4 | 47.7 | 4.4 | 49.8 |
| llama-30B | 4.5 | 58.7 | 2.9 | 55.9 | 3.0 | 61.1 | 4.5 | 58.8 | 3.3 | 59.6 | 2.8 | 60.6 |
| llama-65B | 1.8 | 63.5 | 3.2 | 62.0 | 2.4 | 65.5 | 1.8 | 63.5 | 1.4 | 64.2 | 2.0 | 65.1 |
| llama-2-7B | 12.6 | 44.6 | 4.1 | 45.7 | 4.7 | 50.0 | 7.7 | 45.8 | 6.3 | 47.4 | 5.1 | 49.1 |
| llama-2-13B | 4.1 | 55.5 | 5.6 | 54.7 | 4.2 | 57.2 | 3.6 | 54.5 | 3.6 | 55.4 | 3.9 | 56.6 |
| llama-2-70B | 1.9 | 69.0 | 3.2 | 70.2 | 3.5 | 70.5 | 2.4 | 68.9 | 0.4 | 69.4 | 2.2 | 70.2 |
| llama-2-chat-7B | 11.1 | 46.2 | 3.6 | 49.3 | 4.2 | 50.7 | 5.7 | 47.1 | 4.5 | 48.5 | 4.0 | 49.9 |
| llama-2-chat-13B | 8.4 | 53.8 | 2.8 | 56.5 | 4.4 | 56.9 | 5.7 | 54.4 | 4.9 | 55.2 | 4.2 | 56.4 |
| llama-2-chat-70B | 7.7 | 63.0 | 3.6 | 65.0 | 2.3 | 64.8 | 3.5 | 63.5 | 2.0 | 63.9 | 1.7 | 64.6 |
| vicuna-v1.3-7B | 7.5 | 46.6 | 5.9 | 45.9 | 4.7 | 50.1 | 1.2 | 47.0 | 1.6 | 48.2 | 3.7 | 49.5 |
| vicuna-v1.3-13B | 8.0 | 51.6 | 0.9 | 52.4 | 3.0 | 55.7 | 4.2 | 52.3 | 3.3 | 53.6 | 3.0 | 55.0 |
| vicuna-v1.3-33B | 5.4 | 59.2 | 3.2 | 60.7 | 2.7 | 62.4 | 4.3 | 59.2 | 3.3 | 60.3 | 2.7 | 61.7 |
| vicuna-v1.5-7B | 5.8 | 50.0 | 5.3 | 50.8 | 4.2 | 53.2 | 4.4 | 50.6 | 3.8 | 51.5 | 3.7 | 52.6 |
| vicuna-v1.5-13B | 4.5 | 55.9 | 1.8 | 57.9 | 4.6 | 58.2 | 5.4 | 55.7 | 4.7 | 56.7 | 4.4 | 57.7 |
| falcon-7B | 27.3 | 27.1 | 3.4 | 34.0 | 4.1 | 36.2 | 7.8 | 30.2 | 6.6 | 32.4 | 5.0 | 34.9 |
| falcon-40B | 8.4 | 55.6 | 1.9 | 51.0 | 4.2 | 57.8 | 4.6 | 56.2 | 2.7 | 56.7 | 3.4 | 57.4 |
| falcon-inst-7B | 27.8 | 25.0 | 4.2 | 31.6 | 1.0 | 33.5 | 4.3 | 28.8 | 2.7 | 30.4 | 1.2 | 32.5 |
| falcon-inst-40B | 9.9 | 54.1 | 3.6 | 52.4 | 4.4 | 56.2 | 2.3 | 54.5 | 2.5 | 55.1 | 3.8 | 55.8 |
| gpt-3.5-turbo | 3.8 | 70.9 | 0.7 | 71.8 | 0.5 | 73.3 | 2.1 | 71.1 | 1.1 | 71.9 | 0.3 | 72.8 |
| Δ Average | | | -6.1 | +1.2 | -5.8 | +4.0 | -5.3 | +0.7 | -6.1 | +1.9 | -6.1 | +3.3 |
| ARC | Cost: ×1 | | Cost: ×1 | | Cost: ×4 | | Cost: ×1.15 | | Cost: ×2.2 | | Cost: ×3.4 | |
| llama-7B | 18.8 | 38.8 | 6.7 | 46.2 | 1.9 | 51.6 | 4.7 | 39.6 | 3.3 | 44.1 | 1.9 | 49.5 |
| llama-13B | 12.9 | 55.4 | 1.5 | 56.0 | 2.6 | 62.0 | 10.2 | 56.5 | 6.9 | 58.7 | 3.7 | 60.8 |
| llama-30B | 3.2 | 73.0 | 3.1 | 74.8 | 2.2 | 76.7 | 3.7 | 73.4 | 2.4 | 74.9 | 2.1 | 76.0 |
| llama-65B | 2.1 | 78.5 | 2.3 | 80.2 | 0.9 | 81.8 | 4.0 | 78.4 | 2.6 | 79.8 | 1.3 | 81.4 |
| llama-2-7B | 9.7 | 56.0 | 1.6 | 56.8 | 2.6 | 61.8 | 7.2 | 57.5 | 3.8 | 59.2 | 2.5 | 60.9 |
| llama-2-13B | 4.5 | 66.8 | 6.6 | 68.4 | 2.3 | 69.8 | 4.9 | 67.2 | 3.8 | 68.5 | 2.9 | 69.0 |
| llama-2-70B | 1.4 | 84.4 | 2.0 | 85.4 | 0.6 | 87.2 | 3.1 | 84.1 | 1.8 | 85.3 | 0.7 | 86.3 |
| llama-2-chat-7B | 7.8 | 58.8 | 3.0 | 60.1 | 3.3 | 62.6 | 5.6 | 58.9 | 4.0 | 60.2 | 3.4 | 61.9 |
| llama-2-chat-13B | 9.2 | 68.1 | 8.3 | 70.0 | 3.7 | 71.1 | 4.9 | 68.6 | 3.1 | 69.4 | 3.2 | 70.5 |
| llama-2-chat-70B | 5.8 | 80.0 | 5.3 | 81.4 | 2.8 | 83.9 | 4.3 | 80.9 | 3.4 | 81.8 | 2.8 | 83.3 |
| vicuna-v1.3-7B | 6.0 | 57.4 | 1.7 | 55.9 | 2.4 | 59.3 | 5.0 | 57.3 | 4.1 | 57.8 | 2.9 | 58.9 |
| vicuna-v1.3-13B | 5.6 | 65.8 | 3.9 | 67.0 | 2.2 | 68.4 | 5.4 | 66.0 | 3.1 | 67.2 | 1.4 | 67.6 |
| vicuna-v1.3-33B | 3.8 | 74.7 | 2.4 | 78.0 | 3.5 | 79.2 | 3.2 | 74.9 | 2.4 | 76.1 | 2.5 | 78.1 |
| vicuna-v1.5-7B | 5.2 | 62.4 | 5.5 | 61.6 | 2.3 | 64.9 | 7.0 | 62.9 | 5.1 | 63.7 | 2.9 | 64.5 |
| vicuna-v1.5-13B | 3.7 | 71.6 | 4.7 | 72.8 | 2.5 | 72.6 | 4.8 | 71.4 | 2.9 | 71.9 | 1.5 | 72.6 |
| falcon-7B | 24.0 | 29.5 | 6.1 | 34.7 | 5.3 | 38.3 | 10.5 | 30.0 | 8.9 | 33.0 | 6.6 | 36.4 |
| falcon-40B | 5.3 | 69.9 | 2.8 | 66.4 | 2.1 | 72.5 | 2.0 | 69.2 | 1.7 | 70.8 | 2.0 | 72.0 |
| falcon-inst-7B | 33.2 | 24.4 | 6.4 | 33.5 | 3.4 | 35.3 | 4.6 | 29.3 | 2.0 | 31.4 | 2.1 | 33.7 |
| falcon-inst-40B | 8.9 | 66.2 | 7.1 | 64.3 | 1.9 | 70.4 | 2.9 | 66.8 | 1.8 | 69.8 | 1.6 | 69.9 |
| gpt-3.5-turbo | 1.9 | 86.6 | 0.9 | 86.1 | 1.6 | 88.0 | 1.9 | 86.8 | 1.8 | 87.3 | 1.5 | 87.7 |
| Δ Average | | | -4.6 | +1.6 | -6.1 | +4.5 | -3.7 | +0.7 | -5.2 | +2.1 | -6.2 | +3.6 |
| CSQA | Cost: ×1 | | Cost: ×1 | | Cost: ×5 | | Cost: ×1.2 | | Cost: ×2.6 | | Cost: ×4.2 | |
| llama-7B | 22.4 | 42.3 | 5.0 | 49.8 | 1.4 | 58.5 | 5.6 | 44.6 | 3.8 | 49.4 | 2.2 | 55.4 |
| llama-13B | 13.0 | 62.3 | 3.4 | 66.4 | 2.0 | 68.5 | 6.4 | 62.9 | 4.1 | 64.7 | 2.5 | 67.1 |
| llama-30B | 4.8 | 73.6 | 2.0 | 76.5 | 2.8 | 75.1 | 3.4 | 74.0 | 3.1 | 74.7 | 2.8 | 75.0 |
| llama-65B | 4.5 | 76.2 | 1.9 | 77.9 | 1.3 | 78.7 | 3.6 | 76.1 | 2.7 | 77.1 | 1.8 | 78.3 |
| llama-2-7B | 21.7 | 56.4 | 5.9 | 64.6 | 2.4 | 65.2 | 8.8 | 58.0 | 6.5 | 60.2 | 3.8 | 63.8 |
| llama-2-13B | 7.3 | 68.2 | 2.3 | 70.7 | 4.2 | 71.4 | 5.3 | 68.5 | 4.8 | 69.8 | 4.5 | 71.2 |
| llama-2-70B | 4.2 | 77.9 | 3.3 | 79.8 | 2.3 | 81.6 | 1.9 | 79.3 | 1.3 | 79.8 | 1.7 | 80.8 |
| llama-2-chat-7B | 13.3 | 59.8 | 3.9 | 67.5 | 1.4 | 65.5 | 12.3 | 60.5 | 8.3 | 62.3 | 3.3 | 64.2 |
| llama-2-chat-13B | 3.7 | 68.3 | 3.5 | 72.2 | 3.1 | 71.7 | 6.4 | 69.6 | 4.9 | 70.7 | 3.6 | 71.5 |
| llama-2-chat-70B | 4.9 | 76.7 | 2.2 | 78.0 | 2.0 | 79.9 | 4.7 | 76.8 | 3.1 | 78.1 | 2.0 | 79.5 |
| vicuna-v1.3-7B | 14.4 | 58.5 | 5.0 | 62.1 | 1.7 | 64.7 | 6.3 | 60.2 | 3.9 | 61.8 | 1.4 | 63.7 |
| vicuna-v1.3-13B | 7.5 | 67.1 | 4.4 | 70.3 | 3.8 | 69.9 | 6.8 | 66.7 | 5.2 | 68.0 | 4.1 | 69.0 |
| vicuna-v1.3-33B | 4.2 | 74.8 | 7.4 | 77.1 | 0.7 | 76.4 | 6.0 | 74.3 | 4.4 | 74.7 | 1.7 | 75.9 |
| vicuna-v1.5-7B | 13.0 | 61.7 | 9.0 | 67.1 | 0.8 | 66.4 | 7.8 | 62.9 | 5.2 | 64.1 | 2.4 | 65.8 |
| vicuna-v1.5-13B | 5.7 | 70.6 | 2.5 | 74.3 | 2.0 | 73.8 | 5.4 | 70.4 | 4.5 | 71.8 | 3.0 | 73.2 |
| falcon-7B | 35.7 | 19.8 | 3.2 | 42.2 | 2.3 | 31.1 | 13.1 | 24.7 | 8.4 | 26.7 | 3.2 | 29.1 |
| falcon-40B | 8.0 | 69.1 | 2.6 | 72.1 | 1.0 | 73.8 | 2.5 | 70.6 | 2.2 | 71.9 | 1.4 | 73.4 |
| falcon-inst-7B | 31.4 | 21.1 | 4.0 | 39.6 | 3.5 | 34.7 | 7.1 | 24.5 | 4.5 | 27.9 | 3.0 | 32.6 |
| falcon-inst-40B | 8.3 | 68.1 | 4.9 | 71.1 | 2.2 | 72.2 | 3.8 | 69.7 | 2.9 | 70.5 | 2.5 | 71.9 |
| gpt-3.5-turbo | 4.0 | 79.6 | 2.7 | 82.1 | 1.6 | 82.1 | 2.8 | 80.5 | 2.3 | 80.8 | 1.9 | 81.8 |
| Δ Average | | | -5.5 | +4.0 | -9.5 | +5.5 | -5.6 | +1.1 | -7.3 | +2.7 | -9.0 | +4.6 |

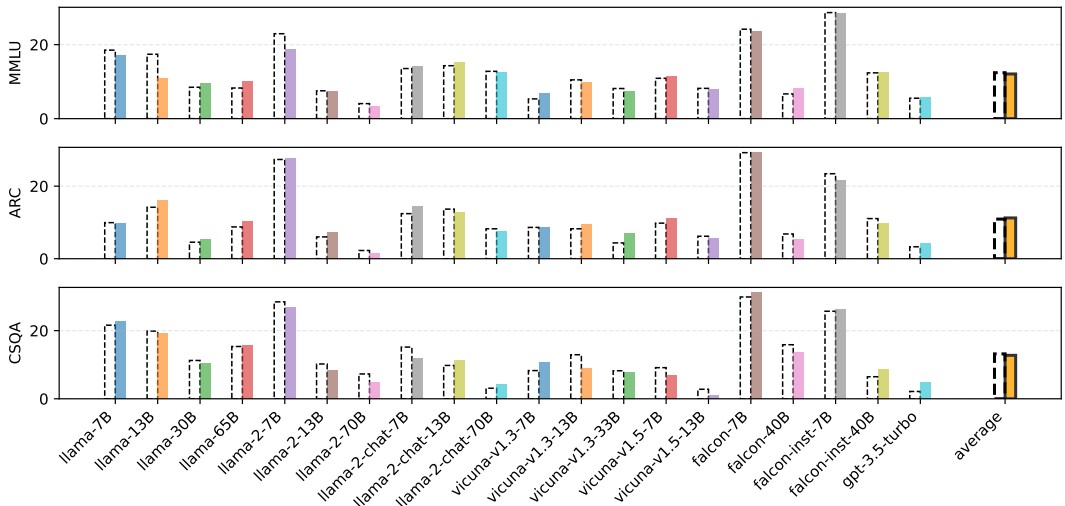

Figure 18: Selection bias before (dashed bars) and after (solid bars) shuffling the default options. We observe no consistent and remarkable changes in the recall standard deviation (Y-axis).

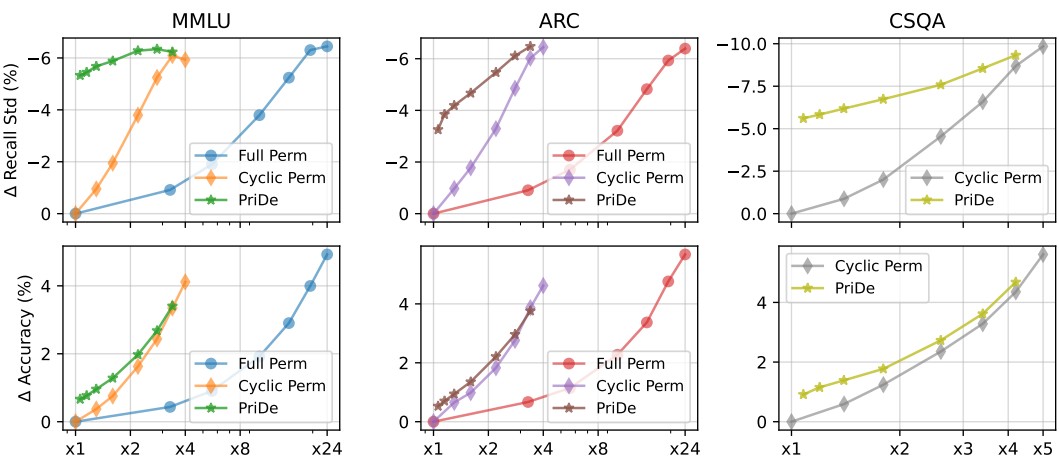

Figure 19: Debiasing results (5-shot) under varying computational costs.

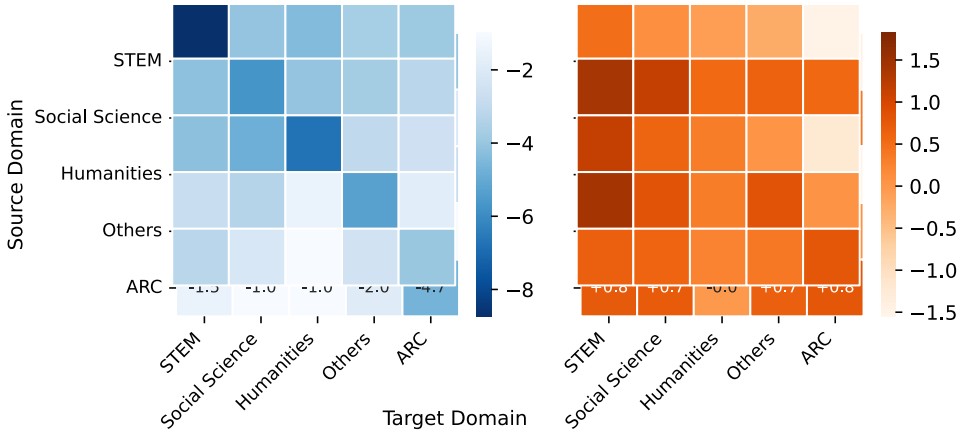

Figure 20: Cross-domain debiasing results (5-shot). We notice slightly degraded transferability under the 5-shot setting, which can be expected due to both the impact of in-context examples themselves and the alteration of in-context examples.

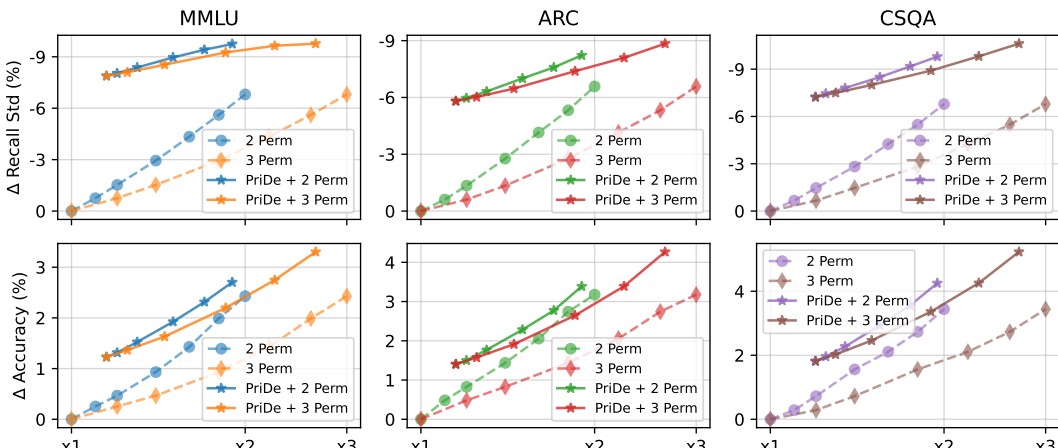

Figure 21: Debiasing results (0-shot) of combining PriDe with two low-cost permutation-based debiasing alternatives **2/3 Perm**. Specifically, we randomly sample another 1 or 2 permutations from the standard Cyclic Perm $\mathcal{I}$, which forms a subset $\widetilde{\mathcal{I}} \subset \mathcal{I}$ containing 2 or 3 permutations in total. For instance, one possible 2-Perm $\widetilde{\mathcal{I}}$ is $\{(0, 1, 2, 3), (1, 2, 3, 0)\}$ and one possible 3-Perm $\widetilde{\mathcal{I}}$ is $\{(0, 1, 2, 3), (2, 3, 0, 1), (3, 0, 1, 2)\}$. In 2/3 Perm, we debias the model prediction with $\widetilde{\mathcal{I}}$ using Equation 1. We control the costs of 2/3 Perm via the ratio $\beta$ of the debiased test samples, where we take $\beta \in \{0\%, 10\%, 20\%, 40\%, 60\%, 80\%, 100\%\}$.

We combine PriDe with 2/3 Perm in three steps: (1) We first estimate the prior $\widetilde{P}_{\mathrm{prior}}$ with 5% test samples $\mathcal{D}_{\mathrm{e}}$ (these samples are meanwhile directly debiased via Cyclic Perm). (2) We then debias $\alpha - 5\%$ samples through 2/3 Perm, with $\widetilde{\mathcal{I}}$ and using Equation 1. (3) We finally debias the remaining $1 - \alpha$ samples $\mathcal{D}_{\mathrm{r}}$ with the estimated prior $\widetilde{P}_{\mathrm{prior}}$, using Equation 8. We take $\alpha \in \{5\%, 10\%, 20\%, 40\%, 60\%, 80\%\}$.

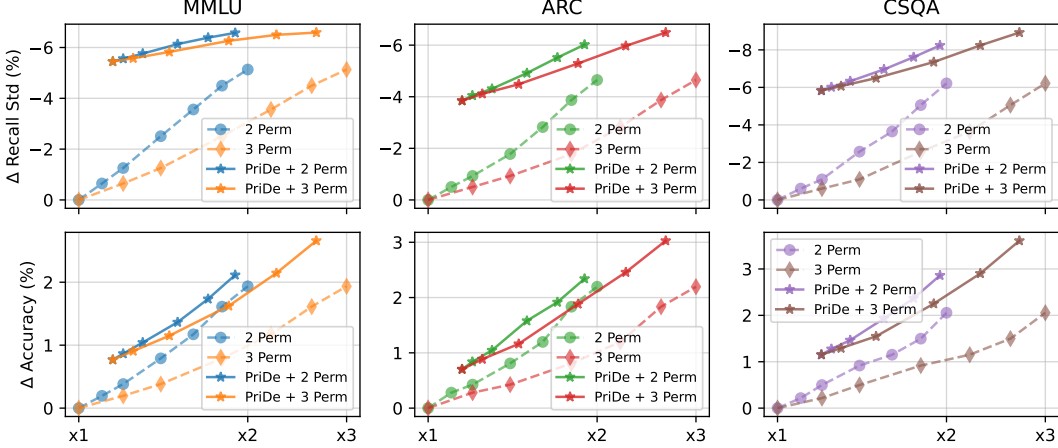

Figure 22: Debiasing results (5-shot) of combining PriDe with 2/3 Perm.

Table 5: Breakdown of the model prediction changes (0-shot MMLU) after **PriDe** (single run, prior estimated with $\alpha = 5\%$ samples and statistics on the remaining $1 - \alpha$ samples) or **Cyclic Permutation** debiasing (statistics on the same samples as PriDe).

**Upper**: We show the proportions of test samples where the model predictions change after debiasing (**Changed**). Among these samples, we count the proportions of predictions changing from wrong to wrong ($\mathbf{F} \rightarrow \mathbf{F}$, but no such samples for all the models), from correct to wrong ($\mathbf{T} \rightarrow \mathbf{F}$), and from wrong to correct ($\mathbf{F} \rightarrow \mathbf{T}$).

**Lower**: We show the prediction probabilities of the predicted answers before and after debiasing, i.e., $\max_i P_{\text{observed}}(d_i|q, x)$ and $\max_i \widetilde{P}_{\text{debiased}}(o_i|q, x)$, averaged over **only prediction-changed** samples or **all** the samples.

| Models | PriDe Debiasing | | | | Cyclic Permutation Debiasing | | | |
|---|---|---|---|---|---|---|---|---|
| | Changed | F→F | T→F | F→T | Changed | F→F | T→F | F→T |
| llama-7B | 27.2% | 43.7% | 25.6% | 30.7% | 50.6% | 38.1% | 21.3% | 40.7% |
| llama-13B | 24.9% | 41.7% | 26.3% | 32.0% | 54.8% | 35.1% | 20.4% | 44.5% |
| llama-30B | 11.8% | 43.4% | 25.2% | 31.4% | 24.6% | 35.0% | 24.2% | 40.8% |
| llama-65B | 17.0% | 39.9% | 26.0% | 34.2% | 29.7% | 33.6% | 25.0% | 41.4% |
| llama-2-7B | 37.2% | 44.5% | 22.2% | 33.3% | 45.6% | 35.7% | 20.2% | 44.1% |
| llama-2-13B | 11.2% | 39.4% | 27.1% | 33.5% | 28.6% | 35.0% | 25.8% | 39.2% |
| llama-2-70B | 9.7% | 37.9% | 30.3% | 31.8% | 17.3% | 27.1% | 28.9% | 44.0% |
| llama-2-chat-7B | 13.1% | 39.6% | 28.9% | 31.5% | 34.4% | 37.0% | 26.6% | 36.4% |
| llama-2-chat-13B | 14.5% | 37.9% | 26.6% | 35.5% | 33.6% | 37.2% | 26.5% | 36.3% |
| llama-2-chat-70B | 5.5% | 29.3% | 28.4% | 42.3% | 22.4% | 29.6% | 29.0% | 41.4% |
| vicuna-v1.3-7B | 15.9% | 46.8% | 25.7% | 27.5% | 33.0% | 41.0% | 25.5% | 33.4% |
| vicuna-v1.3-13B | 17.7% | 42.9% | 27.7% | 29.3% | 32.5% | 37.4% | 26.5% | 36.1% |
| vicuna-v1.3-33B | 4.0% | 39.7% | 26.1% | 34.3% | 23.0% | 35.8% | 24.7% | 39.6% |
| vicuna-v1.5-7B | 13.8% | 43.5% | 27.5% | 29.0% | 33.4% | 38.3% | 27.5% | 34.2% |
| vicuna-v1.5-13B | 12.6% | 37.9% | 27.7% | 34.4% | 27.7% | 39.5% | 25.9% | 34.6% |
| falcon-7B | 50.0% | 48.4% | 23.5% | 28.1% | 65.9% | 43.5% | 21.5% | 34.9% |
| falcon-40B | 10.7% | 39.7% | 28.7% | 31.6% | 33.3% | 34.9% | 27.0% | 38.1% |
| falcon-inst-7B | 46.4% | 47.4% | 23.6% | 29.0% | 71.7% | 44.4% | 21.3% | 34.2% |
| falcon-inst-40B | 16.6% | 36.8% | 30.5% | 32.7% | 34.9% | 37.2% | 28.2% | 34.6% |
| gpt-3.5-turbo | 4.0% | 24.8% | 34.5% | 40.8% | 15.1% | 27.8% | 29.1% | 43.1% |

| Models | Original Prob $\max_i P_{\text{observed}}(d_i|q, x)$ | | | **PriDe**-debiased Prob $\max_i \widetilde{P}_{\text{debiased}}(o_i|q, x)$ | | **Cyclic**-debiased Prob $\max_i \widetilde{P}_{\text{debiased}}(o_i|q, x)$ | |
|---|---|---|---|---|---|---|---|
| | Over **PriDe-Changed** | Over **Cyclic-Changed** | Over **All** | Over **PriDe-Changed** | Over **All** | Over **Cyclic-Changed** | Over **All** |
| llama-7B | 0.289 | 0.306 | 0.314 | 0.283 | 0.304 | 0.255 | 0.257 |
| llama-13B | 0.341 | 0.434 | 0.453 | 0.329 | 0.421 | 0.261 | 0.270 |
| llama-30B | 0.313 | 0.361 | 0.497 | 0.308 | 0.493 | 0.271 | 0.309 |
| llama-65B | 0.332 | 0.358 | 0.514 | 0.328 | 0.514 | 0.271 | 0.315 |
| llama-2-7B | 0.311 | 0.337 | 0.362 | 0.295 | 0.344 | 0.260 | 0.266 |
| llama-2-13B | 0.312 | 0.360 | 0.459 | 0.313 | 0.455 | 0.271 | 0.297 |
| llama-2-70B | 0.345 | 0.411 | 0.612 | 0.353 | 0.617 | 0.286 | 0.345 |
| llama-2-chat-7B | 0.503 | 0.627 | 0.751 | 0.493 | 0.743 | 0.304 | 0.345 |
| llama-2-chat-13B | 0.538 | 0.631 | 0.780 | 0.546 | 0.767 | 0.313 | 0.362 |
| llama-2-chat-70B | 0.496 | 0.658 | 0.810 | 0.483 | 0.795 | 0.317 | 0.378 |
| vicuna-v1.3-7B | 0.339 | 0.425 | 0.578 | 0.337 | 0.579 | 0.285 | 0.326 |
| vicuna-v1.3-13B | 0.421 | 0.510 | 0.683 | 0.398 | 0.677 | 0.295 | 0.347 |
| vicuna-v1.3-33B | 0.388 | 0.562 | 0.756 | 0.382 | 0.757 | 0.304 | 0.372 |
| vicuna-v1.5-7B | 0.436 | 0.537 | 0.691 | 0.412 | 0.676 | 0.299 | 0.345 |
| vicuna-v1.5-13B | 0.450 | 0.529 | 0.715 | 0.423 | 0.703 | 0.307 | 0.365 |
| falcon-7B | 0.299 | 0.312 | 0.319 | 0.286 | 0.294 | 0.251 | 0.251 |
| falcon-40B | 0.339 | 0.417 | 0.556 | 0.345 | 0.553 | 0.280 | 0.320 |
| falcon-inst-7B | 0.376 | 0.413 | 0.419 | 0.330 | 0.341 | 0.255 | 0.255 |
| falcon-inst-40B | 0.387 | 0.455 | 0.593 | 0.366 | 0.577 | 0.284 | 0.326 |
| gpt-3.5-turbo | 0.463 | 0.630 | 0.860 | 0.486 | 0.857 | 0.318 | 0.408 |

Table 6: Breakdown of the model prediction changes (0-shot MMLU) after **PriDe** or **Cyclic Permutation** debiasing (cont.). We count where the debiased/original predicted answers are ranked in the original/debiased prediction distributions, i.e., the ranking of $\arg\max_i \widetilde{P}_{\text{debiased}}(o_i|q,x)$ in $P_{\text{observed}}(d_i|q,x)$ and the ranking of $\arg\max_i P_{\text{observed}}(d_i|q,x)$ in $\widetilde{P}_{\text{debiased}}(o_i|q,x)$. Ranking counting is presented in descending order, i.e., first / second / third / last %.

| Models | PriDe Debiasing | |
|---|---|---|
| | Debiased Prediction $\arg\max_i \widetilde{P}_{\text{debiased}}(o_i\|q,x)$ Ranked in Original Distribution $P_{\text{observed}}(d_i\|q,x)$ | Original Prediction $\arg\max_i P_{\text{observed}}(d_i\|q,x)$ Ranked in Debiased Distribution $\widetilde{P}_{\text{debiased}}(o_i\|q,x)$ |
| `llama-7B` | 72.8% / 18.7% / 6.7% / 1.8% | 72.8% / 22.1% / 5.1% / 0.0% |
| `llama-13B` | 75.1% / 16.7% / 6.6% / 1.7% | 75.1% / 12.8% / 8.9% / 3.2% |
| `llama-30B` | 88.2% / 8.9% / 2.2% / 0.7% | 88.2% / 9.9% / 1.7% / 0.2% |
| `llama-65B` | 83.0% / 11.4% / 4.2% / 1.4% | 83.0% / 12.3% / 3.7% / 1.0% |
| `llama-2-7B` | 62.8% / 17.3% / 14.8% / 5.2% | 62.8% / 24.2% / 11.3% / 1.7% |
| `llama-2-13B` | 88.8% / 10.0% / 1.2% / 0.0% | 88.8% / 8.0% / 3.0% / 0.2% |
| `llama-2-70B` | 90.3% / 8.8% / 0.9% / 0.1% | 90.3% / 7.7% / 1.6% / 0.4% |
| `llama-2-chat-7B` | 86.9% / 10.6% / 2.2% / 0.3% | 86.9% / 10.6% / 2.4% / 0.0% |
| `llama-2-chat-13B` | 85.5% / 12.6% / 1.7% / 0.1% | 85.5% / 10.7% / 3.7% / 0.1% |
| `llama-2-chat-70B` | 94.5% / 5.0% / 0.4% / 0.0% | 94.5% / 4.8% / 0.7% / 0.0% |
| `vicuna-v1.3-7B` | 84.1% / 11.0% / 4.6% / 0.3% | 84.1% / 9.3% / 6.3% / 0.3% |
| `vicuna-v1.3-13B` | 82.3% / 12.2% / 4.6% / 1.0% | 82.3% / 15.4% / 2.0% / 0.4% |
| `vicuna-v1.3-33B` | 96.0% / 3.5% / 0.5% / 0.1% | 96.0% / 3.7% / 0.3% / 0.0% |
| `vicuna-v1.5-7B` | 86.2% / 11.9% / 1.4% / 0.4% | 86.2% / 12.1% / 1.3% / 0.4% |
| `vicuna-v1.5-13B` | 87.4% / 11.2% / 1.3% / 0.1% | 87.4% / 11.9% / 0.6% / 0.0% |
| `falcon-7B` | 50.0% / 29.1% / 14.3% / 6.7% | 50.0% / 18.6% / 11.1% / 20.3% |
| `falcon-40B` | 89.3% / 9.1% / 1.4% / 0.2% | 89.3% / 8.3% / 1.9% / 0.5% |
| `falcon-inst-7B` | 53.6% / 27.8% / 13.7% / 4.9% | 53.6% / 16.9% / 17.2% / 12.3% |
| `falcon-inst-40B` | 83.4% / 11.4% / 4.3% / 0.9% | 83.4% / 13.3% / 3.3% / 0.0% |
| `gpt-3.5-turbo` | 96.0% / 3.8% / 0.2% / 0.0% | 96.1% / 3.5% / 0.4% / 0.0% |

| Models | Cyclic Permutation Debiasing | |
|---|---|---|
| | Debiased Prediction $\arg\max_i \widetilde{P}_{\text{debiased}}(o_i\|q,x)$ Ranked in Original Distribution $P_{\text{observed}}(d_i\|q,x)$ | Original Prediction $\arg\max_i P_{\text{observed}}(d_i\|q,x)$ Ranked in Debiased Distribution $\widetilde{P}_{\text{debiased}}(o_i\|q,x)$ |
| `llama-7B` | 49.4% / 23.8% / 15.2% / 11.6% | 49.4% / 24.5% / 15.1% / 11.0% |
| `llama-13B` | 45.2% / 25.6% / 17.6% / 11.6% | 45.2% / 23.8% / 17.5% / 13.5% |
| `llama-30B` | 75.4% / 16.4% / 5.5% / 2.7% | 75.4% / 15.6% / 6.2% / 2.8% |
| `llama-65B` | 70.3% / 14.6% / 6.7% / 8.3% | 70.3% / 15.0% / 9.5% / 5.2% |
| `llama-2-7B` | 54.4% / 21.5% / 14.4% / 9.8% | 54.4% / 24.1% / 13.5% / 8.0% |
| `llama-2-13B` | 71.4% / 17.9% / 7.5% / 3.2% | 71.4% / 17.6% / 7.8% / 3.3% |
| `llama-2-70B` | 82.7% / 13.2% / 3.1% / 1.0% | 82.7% / 12.1% / 3.9% / 1.3% |
| `llama-2-chat-7B` | 65.6% / 24.4% / 8.0% / 2.0% | 65.6% / 14.1% / 12.4% / 7.9% |
| `llama-2-chat-13B` | 66.4% / 23.5% / 9.0% / 1.1% | 66.4% / 12.1% / 11.4% / 10.1% |
| `llama-2-chat-70B` | 77.6% / 17.6% / 3.9% / 0.9% | 77.6% / 11.9% / 7.5% / 3.0% |
| `vicuna-v1.3-7B` | 67.0% / 17.6% / 11.0% / 4.4% | 67.0% / 15.1% / 7.8% / 10.2% |
| `vicuna-v1.3-13B` | 67.5% / 22.6% / 7.4% / 2.6% | 67.5% / 15.2% / 12.7% / 4.7% |
| `vicuna-v1.3-33B` | 77.0% / 18.2% / 3.6% / 1.1% | 77.0% / 12.6% / 7.7% / 2.7% |
| `vicuna-v1.5-7B` | 66.6% / 20.7% / 10.9% / 1.7% | 66.6% / 13.8% / 7.6% / 11.9% |
| `vicuna-v1.5-13B` | 72.3% / 21.6% / 5.0% / 1.2% | 72.3% / 13.6% / 9.0% / 5.1% |
| `falcon-7B` | 34.1% / 25.0% / 21.3% / 19.6% | 34.1% / 23.9% / 21.2% / 20.8% |
| `falcon-40B` | 66.7% / 19.9% / 11.7% / 1.7% | 66.7% / 16.7% / 8.9% / 7.7% |
| `falcon-inst-7B` | 28.3% / 29.5% / 22.4% / 19.8% | 28.3% / 28.2% / 24.3% / 19.2% |
| `falcon-inst-40B` | 65.1% / 19.3% / 13.4% / 2.2% | 65.1% / 18.0% / 11.8% / 5.1% |
| `gpt-3.5-turbo` | 84.9% / 12.2% / 2.4% / 0.5% | 84.9% / 10.3% / 3.4% / 1.4% |

# D  SUPPLEMENTARY PROOF FOR PERMUTATION-BASED DEBIASING BASELINE

Our probability decomposition assumption (Equation 3) can also give a theoretical proof of the soundness of the permutation-based debiasing baseline (Equation 1). Note that $g_I$ is the inverse mapping of $f_I$: $g_I = f_I^{-1}$, so we can rewrite Equation 3 as:

$$P_{\text{observed}}(d_{g_I(i)}|q, x^I) = Z_{q,x^I}^{-1} P_{\text{prior}}(d_{g_I(i)}|q) P_{\text{debiased}}(o_i|q, x), \;\; \forall I \in \mathcal{I}, i \in \{1, 2, ..., n\}. \quad (9)$$

By taking the logarithm and summing over $I \in \mathcal{I}$, we can similarly derive:

$$\sum_{I \in \mathcal{I}} \log P_{\text{observed}}(d_{g_I(i)}|q, x^I) = \left( \sum_{I \in \mathcal{I}} \log P_{\text{prior}}(d_{g_I(i)}|q) \right) + |\mathcal{I}| \log P_{\text{debiased}}(o_i|q, x) + C \quad (10)$$

$$= \left( \frac{|\mathcal{I}|}{n} \sum_{j=1}^{n} \log P_{\text{prior}}(d_j|q) \right) + |\mathcal{I}| \log P_{\text{debiased}}(o_i|q, x) + C \quad (11)$$

$$= |\mathcal{I}| \log P_{\text{debiased}}(o_i|q, x) + C', \;\; i \in \{1, 2, ..., n\}, \quad (12)$$

and obtain:

$$P_{\text{debiased}}(o_i|q, x) = \text{softmax}\left( \frac{1}{|\mathcal{I}|} \sum_{I \in \mathcal{I}} \log P_{\text{observed}}(d_{g_I(i)}|q, x^I) \right) \quad (13)$$

$$\approx \frac{1}{|\mathcal{I}|} \sum_{I \in \mathcal{I}} P_{\text{observed}}(d_{g_I(i)}|q, x^I) \quad (14)$$

$$= \widetilde{P}_{\text{debiased}}(o_i|q, x), \;\; i \in \{1, 2, ..., n\}, \quad (15)$$

where the approximation holds when the variance of $\{P_{\text{observed}}(d_{g_I(i)}|q, x^I)|I \in \mathcal{I}\}$ is not too large.

# E  EVALUATION DATA AND STATISTICS

**MMLU** `https://github.com/hendrycks/test`

**ARC** `https://allenai.org/data/arc`

**CSQA** `https://allenai.org/data/commonsenseqa`

Table 7: Data statistics of benchmarks used in our experiments. Due to the budget constraints and rate limits, for `gpt-3.5-turbo` in the MMLU evaluation, we randomly sampled 100 test samples from each subject (5,700 in total, accounting for 40% the original size).

$^*$We excluded a few MMLU samples (about 3.2% in total), where the options refer to each other like "A and B", "none of the above", in all the experiments. The exclusion of these samples is to ensure the validity of option position changes in the "answer-moving attack" and option shuffling or permutation.

| Benchmarks | | # Samples | # Evaluated$^*$ | # Options | Golden Answer Distribution |
|---|---|---|---|---|---|
| MMLU | STEM | 3,018 | 2,925 | | 21.4% / 23.8% / 25.9% / 28.9% |
| | Social Science | 3,077 | 2,992 | 4 | 21.7% / 23.4% / 23.8% / 31.1% |
| | Humanities | 4,705 | 4,476 | | 24.2% / 24.5% / 27.1% / 24.2% |
| | Others | 3,242 | 3,199 | | 23.8% / 26.8% / 24.4% / 25.0% |
| | Overall | 14,042 | 13,592 | 4 | 22.9% / 24.7% / 25.5% / 26.9% |
| ARC | | 1,165 | 1,165 | 4 | 22.6% / 26.5% / 26.5% / 24.4% |
| CSQA | | 1,216 | 1,216 | 5 | 19.6% / 20.8% / 19.8% / 20.6% / 19.2% |

## F    REFERENCE PROJECTS

**HuggingFace LLM Leaderboard**    `https://huggingface.co/spaces/HuggingFaceH4/open_llm_leaderboard`

**EleutherAI lm-harness** `https://github.com/EleutherAI/lm-evaluation-harness`

**Original MMLU implementation** `https://github.com/hendrycks/test`

**OpenAI Evals** `https://github.com/openai/evals`

## G    EVALUATED OPEN-SOURCE MODELS

| Models | URLs |
|---|---|
| llama-7b | https://huggingface.co/huggyllama/llama-7b |
| llama-13b | https://huggingface.co/huggyllama/llama-13b |
| llama-30b | https://huggingface.co/huggyllama/llama-30b |
| llama-65b | https://huggingface.co/huggyllama/llama-65b |
| llama-2-7b | https://huggingface.co/meta-llama/Llama-2-7b-hf |
| llama-2-13b | https://huggingface.co/meta-llama/Llama-2-13b-hf |
| llama-2-70b | https://huggingface.co/meta-llama/Llama-2-70b-hf |
| llama-2-chat-7b | https://huggingface.co/meta-llama/Llama-2-7b-chat-hf |
| llama-2-chat-13b | https://huggingface.co/meta-llama/Llama-2-13b-chat-hf |
| llama-2-chat-70b | https://huggingface.co/meta-llama/Llama-2-70b-chat-hf |
| vicuna-v1.3-7b | https://huggingface.co/lmsys/vicuna-7b-v1.3 |
| vicuna-v1.3-13b | https://huggingface.co/lmsys/vicuna-13b-v1.3 |
| vicuna-v1.3-33b | https://huggingface.co/lmsys/vicuna-33b-v1.3 |
| vicuna-v1.5-7b | https://huggingface.co/lmsys/vicuna-7b-v1.5 |
| vicuna-v1.5-13b | https://huggingface.co/lmsys/vicuna-13b-v1.5 |
| falcon-7b | https://huggingface.co/tiiuae/falcon-7b |
| falcon-40b | https://huggingface.co/tiiuae/falcon-40b |
| falcon-inst-7b | https://huggingface.co/tiiuae/falcon-7b-instruct |
| falcon-inst-40b | https://huggingface.co/tiiuae/falcon-7b-instruct |

