# OpenReview forum: "Large Language Models Are Not Robust Multiple Choice Selectors"
_ICLR.cc/2024/Conference — ICLR 2024 spotlight_

### Official Review · Reviewer_VDjF · 2023-10-22

**Soundness:** 3 good
**Presentation:** 2 fair
**Contribution:** 2 fair
**Rating:** 5
**Confidence:** 4

**Summary:**

The paper studies the issue of sensitivity to answer option order in large language models (LLMs), which can lead to biased predictions. It introduces a new method called PriDe to mitigate this sensitivity by estimating and correcting for the model's bias during inference. The results show that PriDe can reduce the prediction sensitivity.

**Strengths:**

- This paper studies the LLMs' sensitivity to order of answer options, which is an important problem in current LLM evaluation, and provides empirical analysis of the underlying reasons.
- The proposed method PriDe operates on test time without introducing extra computational cost, which is suitable for current LLMs.
- The authors conduct extensive experiments including different models, tasks, ablation studies, cross-task evaluation, etc.

**Weaknesses:**

- The proposed method requires sampling test samples first to estimate the prior, which may introduce another dimension of sensitivity of the selection of the test samples. The accuracy of this estimation might vary based on the quality and representativeness of these samples.
- I understand the procedure of cyclic permutation and full permutation, but how are they used as the debiasing methods? Do the authors take the best result of the permutations as the prediction?
- The authors use the balance of recalls and Rstd as the major metrics throughout the paper. Can the authors formally define this? I didn't immediately get it.
- The writing and presentation need more improvement, e.g., I think the proposed PriDe is quite intuitive but the authors introduce too many unnecessary notations ($d_i, o_i, g_i, x_i, .... $) before getting into the real introduction of the method, which makes the reading difficult.

**Questions:**

- The proposed method basically follows estimate-then-mitigate, which is somewhat similar to the calibrate-before-use (Zhao et al. ICML 2021) paper, though this one targets a different setting and is not directly applicable to MCQs. But it would be interesting to compare the differences and know if calibrate-before-use can also help with MCQ sensitivity.

---

> ### Author Response · Authors · 2023-11-13
> **Response to Reviewer VDjF (1/2)**
>
> Thanks for your constructive comments! We address your concerns or questions as follows.
>
> > The proposed method requires sampling test samples first to estimate the prior, which may introduce another dimension of sensitivity of the selection of the test samples. The accuracy of this estimation might vary based on the quality and representativeness of these samples.
>
> Good question! While we do test sample sampling mainly out of the need of experiments, in practice we usually expect the estimated prior to be less sensitive to estimation sample selection. We here supplement statistics in the 5 runs of PriDe (0-shot, $\alpha=5\%$). We present the mean (as reported in our paper) ∆ RStd and ∆ Acc as well as their best/worst values (averaged over all the 20 LLMs):
>
> | Benchmarks | ∆ RStd (mean) | ∆ RStd (best ↓) | ∆ RStd (worst ↑) | ∆ Acc (mean) | ∆ Acc (best ↑) | ∆ Acc (worst ↓) |
> | ---------- | ------------- | --------------- | ---------------- | ------------ | -------------- | --------------- |
> | MMLU       | -7.6          | -8.0            | -7.3             | 1.2          | 1.3            | 1.1             |
> | ARC        | -5.6          | -6.8            | -4.3             | 1.3          | 1.7            | 0.8             |
> | CSQA       | -6.9          | -7.7            | -6.0             | 1.7          | 2.2            | 1.3             |
>
> We observe that the selection of estimation samples may introduce slight fluctuations in debiasing results, and even in the worst-case scenario, PriDe still leads to a notable debiasing performance (decrease in RStd and increase in Acc). Therefore, we believe PriDe's sensitivity to the selection of estimation samples lies within an acceptable range and does not obscure its merit (effectiveness and efficiency).
>
> > I understand the procedure of cyclic permutation and full permutation, but how are they used as the debiasing methods? Do the authors take the best result of the permutations as the prediction?
>
> Cyclic and Full Permutation can be viewed as having a debiasing effect, as they involve swapping options and averaging prediction distributions over different permutations. They can intuitively mitigate the model's bias for option IDs or options' ordering positions. This is similarly done in recent work [1] and [2], where they swap candidate responses to mitigate GPT-4's evaluation bias.
>
> For Full Permutation, there is only one possible permutation set (i.e., all possible permutations). For cyclic permutations, there might be multiple possible permutation sets (as long as we ensure one pairing between each option ID and option content). The selection of cyclic permutation sets is not our focus, as our method PriDe can be directly combined with any reasonable cyclic permutation set. In the main text, we use the simplest and most intuitive set for Cyclic Permutation, e.g., $\{ (1,2,3,4), (2,3,4,1), (3,4,1,2), (4,1,2,3) \}$ for 4-option MCQ tasks. We show in **Section 3.1 and Figure 16 in Appendix F** that selecting other cyclic permutation sets leads to similar debiasing results.
>
> [1] Wang, Peiyi, et al. "Large language models are not fair evaluators." *arXiv preprint arXiv:2305.17926* (2023).
>
> [2] Zheng, Lianmin, et al. "Judging LLM-as-a-judge with MT-Bench and Chatbot Arena." *arXiv preprint arXiv:2306.05685* (2023).
>
> > The authors use the balance of recalls and Rstd as the major metrics throughout the paper. Can the authors formally define this? I didn't immediately get it.
>
> Sure! The recall of an option ID $d_i$ is defined as:
> $$
> \mathrm{Recall}(d_i) = \frac{ \\# (\text{correct answer is } d_i \ \\&\text{ prediction is } d_i) }{ \\# (\text{correct answer is } d_i)} \times 100 \\%,
> $$
> while RStd (Std of recalls) is:
> $$
> \mathrm{RStd}=\mathrm{Std}( \\{ \mathrm{Recall}(d_i) \\}\_{i=1}^n) = \sqrt{\frac{\sum_{i=1}^n (\mathrm{Recall}(d_i) - \mu)^2}{n}}, \text{where } \mu = \frac{1}{n} \sum_{i=1}^n \mathrm{Recall}(d_i).
> $$
> Our motivation of using this measurement is illustrated in Section 2.2.

---

> ### Author Response · Authors · 2023-11-13
> **Response to Reviewer VDjF (2/2)**
>
> > The writing and presentation need more improvement, e.g., I think the proposed PriDe is quite intuitive but the authors introduce too many unnecessary notations ($d_i, o_i, g_i, x_i, .... $) before getting into the real introduction of the method, which makes the reading difficult.
>
> We would like to clarify the reasons for introducing these formal notations before the proposed method. We attempted to introduce the proposed method first and then intersperse or supplement the introduction of the permutation-based baseline on which our method relies. However, we found that this compromised the integrity of the writing content. We also found that without introducing these formal notations, the writing would become very repetitive and lengthy (we would have to repeatedly use the same terms to avoid ambiguity). Additionally, we believed that formal notations could aid in deriving general solution forms. All these above led us to adopt the current formal notations and writing logic.
>
> If you believe there is a better way to present or structure the content, we would greatly appreciate it and be open to taking your suggestions!
>
> > The proposed method basically follows estimate-then-mitigate, which is somewhat similar to the calibrate-before-use (Zhao et al. ICML 2021) paper, though this one targets a different setting and is not directly applicable to MCQs. But it would be interesting to compare the differences and know if calibrate-before-use can also help with MCQ sensitivity.
>
> We are willing to discuss the difference from Contextual Calibration (Zhao et al.). We make a preliminary attempt to adapt Contextual Calibration to MCQ debiasing as follows:
>
> 1. For each test sample, we use the default input and obtain the prediction distribution $\mathbf{p}$.
> 2. We then replace all the options with the same content-free text: the null string `''`, `N/A`, or `[MASK]`, as in Zhao et al., and estimate the model's prediction distribution over the option IDs, denoted as $\mathbf{p}_0$ (we use all the content-free texts and take the average of their $\mathbf{p}_0$).
> 3. We use $\mathbf{p}/\mathbf{p}_0$ after normalization as the "calibrated" prediction distribution, as done in Zhao et al.
>
> So, from the perspective of implementation, Contextual Calibration is similar to PriDe. The key difference lies in how we estimate $\mathbf{p}_0$, which we refer to as "prior" in our work. The results are shown below (`gpt-3.5-turbo-0613`, 0-shot ARC, for a quick verification):
>
> | Methods                | RStd | Acc |
> | ---------------------- | ------ | ----- |
> | Default                | 3.3    | 84.3  |
> | PriDe ($\alpha=5\\%$)   | 2.3    | 84.2  |
> | Contextual Calibration | 4.8    | 83.1  |
>
> We find that Contextual Calibration fails to mitigate selection bias (RStd) and may impair model performance (Acc). It implies that the "prior" ($\mathbf{p}_0$) estimated by Contextual Calibration cannot reflect the model's selection bias in MCQs and may also be hard to interpret.

---

> ### Author Response · Authors · 2023-11-22
> **A Gentle Reminder of the Final Feedback**
>
> Dear Reviewer VDjF,
>
> We would like to thank you for your time and comments. We hope our previous response has adequately resolved your questions or concerns. As the deadline for the ICLR rebuttal period is approaching, we look forward to hearing your feedback on our response, and would be pleased to clarify any additional questions.
>
> Best,
>
> Authors

---

### Official Review · Reviewer_G9aZ · 2023-10-29

**Soundness:** 4 excellent
**Presentation:** 4 excellent
**Contribution:** 3 good
**Rating:** 8
**Confidence:** 3

**Summary:**

This work presents a comprehensive analysis of the selection bias issue in large language models (LLMs) when dealing with multiple choice questions (MCQs).
The experimental results identify the root cause of this bias as the LLMs' token bias, which leads to a preference for specific option IDs when predicting answers.
Based on these observations, this work proposes a label-free, inference-time debiasing method called PriDe, which effectively mitigates selection bias.

**Strengths:**

1. The empirical analysis is thorough, involving 20 LLMs and three benchmark datasets. This extensive evaluation provides strong evidence for the existence of selection bias in LLMs and its impact on their performance in MCQ tasks. The identification of token bias as the primary source of this issue is a valuable insight that can inform future research on LLMs and their limitations.

2. The proposed PriDe method is effective when the computing cost is limited. Further analysis on generalizability reveals that the prior estimated by PriDe can be generalized across tasks.

**Weaknesses:**

1. It seems that PriDe achieves comparable performance with simple baselines when the computation cost is not limitated. In application scenarios, we always first estimate the prior without concerning the computation cost, then apply this prior to serve applications.
It would be better if PriDe could have a higher upper boudn performance.

**Questions:**

1. The generalization analysis indicates that the bias for a certain model is consistent across different tasks.
Could you further demonstrate this with more statics or results?
It would also help to enhace the claimed interpretability.

---

> ### Author Response · Authors · 2023-11-13
> **Response to Reviewer G9aZ**
>
> Thanks for your positive comments! We address your concerns or questions as follows.
>
> > It seems that PriDe achieves comparable performance with simple baselines when the computation cost is not limitated. In application scenarios, we always first estimate the prior without concerning the computation cost, then apply this prior to serve applications. It would be better if PriDe could have a higher upper boudn performance.
>
> (This question is similar to the one raised by Reviewer CGVT, so we use the same answer)
>
> When the budget is sufficient, using more permutations does yield better debiasing effects and performance improvements. As discussed in Section 4.3, this is akin to "mixture of experts" or "model ensemble". Our method, on the other hand, provides a computation-efficient alternative. We believe this could be beneficial for debiasing in scenarios with constrained/limited computational resources, such as platforms like the HuggingFace LLM Leaderboard, where a large number of models need to be evaluated on numerous benchmarks.
>
> > The generalization analysis indicates that the bias for a certain model is consistent across different tasks. Could you further demonstrate this with more statics or results? It would also help to enhace the claimed interpretability.
>
> Of course! Here we compute the L1 distance (due to its intuitiveness, as in our response to Reviewer CGVT) between the estimated priors from different domains to illustrate PriDe's cross-domain generalization (0-shot, averaged over all the LLMs, $\alpha=5\%$; priors' L1 distance is averaged over 5 runs).
>
> | Domain 1 \ Domain 2 | STEM  | Social Science | Humanities | Others | ARC   |
> | ------------------- | ----- | -------------- | ---------- | ------ | ----- |
> | **STEM**            | 0     | 0.104          | 0.094      | 0.106  | 0.121 |
> | **Social Science**  | 0.104 | 0              | 0.099      | 0.067  | 0.076 |
> | **Humanities**      | 0.094 | 0.099          | 0          | 0.110  | 0.125 |
> | **Others**          | 0.106 | 0.067          | 0.110      | 0      | 0.087 |
> | **ARC**             | 0.121 | 0.076          | 0.125      | 0.087  | 0     |
>
> We think these priors' gaps are usually marginal, which could verify that for a certain model, its prior for option IDs is similar across domains.

---

> ### Author Response · Authors · 2023-11-22
> **A Gentle Reminder of the Final Feedback**
>
> Dear Reviewer G9aZ,
>
> We would like to thank you for your time and comments. We hope our previous response has adequately resolved your questions or concerns. As the deadline for the ICLR rebuttal period is approaching, we look forward to hearing your feedback on our response, and would be pleased to clarify any additional questions.
>
> Best,
>
> Authors

---

> > ### Comment · Reviewer_G9aZ · 2023-11-22
> > **Thanks for your response**
> >
> > I have no more questions. I have updated my score.

---

### Official Review · Reviewer_rhkS · 2023-10-31

**Soundness:** 3 good
**Presentation:** 3 good
**Contribution:** 3 good
**Rating:** 8
**Confidence:** 2

**Summary:**

This paper experimentally discovers an issue that LLMs are vulnerable to option position changes, or the Option-Order Sensitivity problem, in MCQs due to their inherent “selection bias.” It proposes a label-free, inference-time debiasing method(PriDe) to mitigate the selection bias. The experimental results demonstrate the claim and the usefulness of the PriDe.

**Strengths:**

I really appreciate the paper conducted extensive experiments to demonstrate and analyze the Option-Order Sensitivity problem. Some observations are really interesting; for example, even the same models with different parameter sizes but trained using the same data exhibit different position preferences.

The PriDe is intuitive but also effective.

**Weaknesses:**

It would be better to cite "Leveraging large language models for multiple choice question answering" or other related papers when mentioning the Option-Order Sensitivity problem since they have found the problem earlier than the work of this paper.

It would be better to analyze more technicals, including self-consistency.

**Questions:**

Please refer to the weakness.

---

> ### Author Response · Authors · 2023-11-13
> **Response to Reviewer rhkS**
>
> Thanks for your positive comments! We address your concerns or questions as follows.
>
> > It would be better to cite "Leveraging large language models for multiple choice question answering" or other related papers when mentioning the Option-Order Sensitivity problem since they have found the problem earlier than the work of this paper.
>
> We appreciate your suggestion! We will add the citation in the revision.
>
> > It would be better to analyze more technicals, including self-consistency.
>
> In Section 2.6, we experimented with simple prompting strategies, considering their popularity in recent research, to observe whether they have a positive impact on debiasing (finding that they do not). We did not explore too much into prompting engineering, for the following reasons:
>
> 1. Our empirical analysis in Sections 2.3-2.5 cannot motivate us to work on prompting engineering, i.e., we intuitively believe that prompting engineering is not the fundamental means of debiasing (and may also be tricky).
> 2. Complex prompting strategies (such as self-consistency) are designed primarily to enhance model performance rather than to debias. Moreover, they typically rely on powerful but often commercial, closed-source LLMs like ChatGPT, Claude, and PaLM, making them less applicable to open-source LLMs like LLaMA.
> 3. Complex prompting strategies are often expensive, especially when involving much sampling or heuristic filtering.
>
> We also supplement the results of Self-Consistency on ARC (this benchmark has a small scale, suitable for quick verification). We employ `gpt-3.5-turbo-0613`, sample 10 Chain-of-Thought paths, and then vote on the predicted results.
>
> | Methods          | RStd ↓ | Acc ↑ |
> | ---------------- | ------ | ----- |
> | Default          | 3.3    | 84.3  |
> | Removing IDs     | 0.6    | 84.9  |
> | Chain-of-Thought | 3.4    | 84.5  |
> | Self-Consistency | 4.5    | 88.9  |
>
> As expected, Self-Consistency improved Acc. However, like other prompting strategies, it cannot mitigate selection bias and even somewhat amplifies it (RStd increases), which is inconsistent with our goal of debiasing. We believe that investigating the impact of prompting strategies on LLMs' behavioral bias would be an intriguing research problem.

---

> ### Author Response · Authors · 2023-11-22
> **A Gentle Reminder of the Final Feedback**
>
> Dear Reviewer rhkS,
>
> We would like to thank you for your time and comments. We hope our previous response has adequately resolved your questions or concerns. As the deadline for the ICLR rebuttal period is approaching, we look forward to hearing your feedback on our response, and would be pleased to clarify any additional questions.
>
> Best,
>
> Authors

---

### Official Review · Reviewer_CGVT · 2023-11-01

**Soundness:** 4 excellent
**Presentation:** 4 excellent
**Contribution:** 3 good
**Rating:** 8
**Confidence:** 4

**Summary:**

This paper investigated the LLMs' sensitivity to position changes in multiple-choice questions, discovered that token bias is the main cause/ Furthermore, the authors proposed a way to efficiently suppress this bias and improve accuracy.

**Strengths:**

1. It flows! The writing is perfect. All sections follow each other naturally, from problem to observation, to diagnosis, to ruling out simplistic solutions, to proposed solutions. In each step, there are corresponding experiments to substantiate it.
2. There are some clever experiment designs in diagnosing the cause, and the experiments are carried out with caution (e.g. replacing symbols to confirm).
3. Comprehensive experiments on many models and datasets.

**Weaknesses:**

1. When the compute budget is unbounded, the proposed method sometimes has a slight accuracy disadvantage compared to full perm.

**Questions:**

1. In deriving the method, there are a few key assumptions, e.g. Prior for option IDs depends mostly on q. Is it possible to empirically verify this assumption?

---

> ### Author Response · Authors · 2023-11-13
> **Response to Reviewer CGVT**
>
> Thanks for your positive comments! We address your concerns or questions as follows.
>
> > When the compute budget is unbounded, the proposed method sometimes has a slight accuracy disadvantage compared to full perm.
>
> (This question is similar to the one raised by Reviewer G9aZ, so we use the same answer)
>
> When the budget is sufficient, using more permutations does yield better debiasing effects and performance improvements. As discussed in Section 4.3, this is akin to "mixture of experts" or "model ensemble". Our method, on the other hand, provides a computation-efficient alternative. We believe this could be beneficial for debiasing in scenarios with constrained/limited computational resources, such as platforms like the HuggingFace LLM Leaderboard, where a large number of models need to be evaluated on numerous benchmarks.
>
> > In deriving the method, there are a few key assumptions, e.g. Prior for option IDs depends mostly on q. Is it possible to empirically verify this assumption?
>
> We removed the dependency on $x^I$ in $P_\textrm{prior}$ because it could be a minimally strong assumption necessary for our derivation. We are also pleased to empirically verify this assumption, that is, whether swapping options (i.e., different $x^I$ w.r.t. $I$) would change the derived $P_\textrm{prior}$? If the answer is no, then our assumption makes sense.
>
> In our main text, we use the cyclic permutation set $\{ (1,2,3,4), (2,3,4,1), (3,4,1,2), (4,1,2,3) \}$ for 4-option MCQ tasks. Our verification contains the following steps:
>
> 1. Modify the default-ordered options as $(1,2,4,3)$ or $(4,3,2,1)$.
> 2. Use the corresponding cyclic set $\{ (1,2,4,3), (2,4,3,1), (4,3,1,2), (3,1,2,4) \}$ or $\{ (4,3,2,1), (3,2,1,4), (2,1,4,3), (1,4,3,2) \}$ to derive $P_\textrm{prior}'$.
> 3. Check if $P_\textrm{prior}'$ is close to $P_\textrm{prior}$. We use the L1 distance as measurement (averaged over all the test samples), due to its intuitiveness: $ d(\mathbf{p}, \mathbf{q})= \sum_i |p_i - q_i|. $
>
> | Models      | MMLU $(1,2,4,3)$ | ARC $(1,2,4,3)$ | MMLU $(4,3,2,1)$ | ARC $(4,3,2,1)$ |
> | ----------- | ---------------- | --------------- | ---------------- | --------------- |
> | llama-7B    | 0.014            | 0.014           | 0.013            | 0.013           |
> | llama-13B   | 0.034            | 0.047           | 0.035            | 0.045           |
> | llama-30B   | 0.068            | 0.079           | 0.066            | 0.077           |
> | llama-65B   | 0.069            | 0.081           | 0.070            | 0.082           |
> | llama-2-7B  | 0.028            | 0.022           | 0.028            | 0.023           |
> | llama-2-13B | 0.061            | 0.059           | 0.058            | 0.056           |
> | llama-2-70B | 0.095            | 0.106           | 0.096            | 0.109           |
> | **Average** | **0.053**        | **0.058**       | **0.052**        | **0.058**       |
>
> We can see that the difference between $P_\textrm{prior}'$ (i.e., estimated with a different permutation set) and $P_\textrm{prior}$ is marginal (their L1 distance is quite small), which we believe could validate the soundness of our assumption.

---

> ### Author Response · Authors · 2023-11-22
> **A Gentle Reminder of the Final Feedback**
>
> Dear Reviewer CGVT,
>
> We would like to thank you for your time and comments. We hope our previous response has adequately resolved your questions or concerns. As the deadline for the ICLR rebuttal period is approaching, we look forward to hearing your feedback on our response, and would be pleased to clarify any additional questions.
>
> Best,
>
> Authors

---

### Author Response · Authors · 2023-11-21
**A Gentle Reminder of the Final Feedback**

We sincerely appreciate the thoughtful comments and constructive feedback of all the reviewers. We are encouraged that the reviewers found:

1. Our paper targets an important research problem (VDjF) and provides interesting and insightful findings (rhkS, G9aZ, VDjF).
2. Our paper writing is smooth and natural (CGVT, "**It flows! The writing is perfect**").
3. Our comprehensive, thorough, and careful evaluation yields convincing empirical observations and results (CGVT, rhkS, G9aZ, VDjF; **all the reviewers!**).
4. Our proposed debiasing method PriDe is effective and efficient at the low computational cost setting (rhkS, G9aZ), and its operation on inference time is suitable for modern LLMs (VDjF).

We hope our responses to the reviewers can adequately resolve your questions or concerns. As the deadline for the ICLR rebuttal period is approaching, we look forward to hearing your feedback on our responses, and would be pleased to clarify any additional questions.

---

### Meta-Review · Area_Chair_RVRA · 2023-12-11

**Metareview:**

This paper conducts an interesting study that shows the significance of the position of options in MCQs when using LLMs for such tasks. The paper comes with an extensive set of analyses and proposes effective methods for mitigating the identified bias. Most reviewers agree the work is making substantial contributions to this specific domain and the results are worth sharing with the community, and may inspire others working on similar or related directions.

**Justification For Why Not Higher Score:**

Although the scores were high, I found the paper in its current form may not yet have significant impact that can be worth sharing with a wider community.

**Justification For Why Not Lower Score:**

This is a great paper, which should be presented orally.

---

### Decision · Program_Chairs · 2024-01-16

Accept (spotlight)